# VAMO: Efficient Zeroth-Order Variance Reduction for SGD with Faster Convergence

## Abstract

Optimizing large-scale nonconvex problems, common in deep learning, demands balancing rapid convergence with computational efficiency. First-order (FO) optimizers, which serve as today's baselines, provide fast convergence and good generalization but often incur high computation and memory costs due to the large size of modern models. Conversely, zeroth-order (ZO) algorithms reduce this burden using estimated gradients, yet their slow convergence in high-dimensional settings limits practicality. We introduce VAMO (VAriance-reduced Mixed-gradient Optimizer), a stochastic variance-reduced method that extends mini-batch SGD with full-batch ZO gradients under an SVRG-style framework. VAMO's hybrid design utilizes a two-point ZO estimator to achieve a dimension-agnostic convergence rate of $\mathcal{O}(1/T + 1/b)$, where $T$ is the number of iterations and $b$ is the batch-size, surpassing the dimension-dependent slowdown of purely ZO methods and significantly improving over SGD's $\mathcal{O}(1/\sqrt{T})$ rate. Additionally, we propose a multi-point variant that mitigates the $O(1/b)$ error by adjusting the number of estimation points to balance convergence and cost. Importantly, VAMO achieves these gains with smaller dynamic memory requirements than many FO baselines, making it particularly attractive for edge deployment. Experiments including traditional neural network training and LLM finetuning confirm that VAMO not only outperforms established FO and ZO methods, but also does so with a light memory footprint.

## 1 Introduction

First-order (FO) optimization methods, particularly Stochastic Gradient Descent (SGD), have been applied in training a wide range of machine learning models. For large-scale problems, variance-reduced (VR) techniques, such as the Stochastic Variance Reduced Gradient (SVRG) algorithm (Johnson & Zhang, 2013; Allen-Zhu & Yuan, 2016; Reddi et al., 2016), offer significant improvements, achieving faster convergence rates, $O(1/T)$, compared to the rate of SGD $O(1/\sqrt{T})$ (Reddi et al., 2016). However, in recent years, extremely large models such as Large Language Models (LLMs) with billions of parameters have become increasingly prevalent in machine learning. When training these models, traditional variance reduction methods like SVRG face a major challenge: they require periodically computing the full gradient over the entire dataset, which requires high cost for such large-scale models (Reddi et al., 2016). For LLMs, this step results in prohibitive computational and memory overhead, severely hindering efficient training of large models.

Zeroth-order (ZO) optimization methods present an appealing alternative in this context, as they completely bypass the need for explicit gradient calculations, estimating gradients using only function value queries (Nesterov & Spokoiny, 2017; Liu et al., 2018b). This gradient-free characteristic drastically reduces per-iteration computational cost and memory footprint, making ZO methods attractive for resource-constrained training of LLMs (Malladi et al., 2023; Gautam et al., 2024). Despite these advantages, ZO methods typically exhibit slower theoretical convergence rates than FO methods and, critically, often suffer from a strong dependence on the parameter dimension $d$ (Duchi et al., 2015; Nesterov & Spokoiny, 2017). Given the vast dimensionality of modern LLMs, this dependence can render pure ZO approaches impractically slow. This creates a clear dilemma for large model training: FO methods offer desirable convergence, but suffer from high gradient costs, while ZO methods are cheaper per step but often too slow and scale poorly with the parameter dimension. This naturally raises the question: can we devise a hybrid strategy that combines the strengths of both FO and ZO methods, thereby overcoming their limitations for efficient training of large models?

In this work, we propose **VAMO (VAriance-reduced Mixed-gradient Optimizer)**, a new adaptive hybrid algorithm specifically designed to navigate this dilemma in large-scale non-convex optimization. Our approach integrates FO and ZO techniques within the SVRG framework, aiming to maintain SVRG's fast convergence while substantially mitigating its computational burden. A major breakthrough here is the replacement of the prohibitively expensive full FO gradient $\nabla f(\hat{\mathbf{x}})$ in the snapshot point with an efficient ZO gradient estimate $\hat{\nabla} f(\hat{\mathbf{x}})$, which significantly reduces the computation. This leads to a convergence rate of $O(1/T + 1/b)$, significantly outperforming FO-SGD, matching the rate of FO-SVRG only with an additional error of $O(1/b)$. The adaptability of VAMO is then enhanced through several key innovations proposed in this work. First, we extend VAMO with a multi-point ZO gradient estimator for the full ZO gradient $\hat{\nabla} f(\hat{\mathbf{x}})$. Compared to the standard two-point variant, this extension reduces the additional $O(1/b)$ error term and allows flexibility in balancing performance and cost by varying the number of query directions. Second, we introduce a mixing coefficient $\alpha$ into the update rule. Though incorporating the full ZO gradient can effectively reduce variance, it inevitably introduces estimation error; the coefficient $\alpha$ provides fine-grained control over this balance by adaptively weighting the FO stochastic gradient and the ZO correction term. Together, these mechanisms make VAMO highly adaptable across diverse optimization scenarios. Importantly, despite this hybrid design, our gradient estimator preserves the unbiasedness of FO-SVRG, distinguishing VAMO from many biased ZO methods and enabling a rigorous convergence analysis with stronger theoretical guarantees.

## 2 RELATED WORK

**First-order optimization.** While Stochastic Gradient Descent (SGD) (Robbins & Monro, 1951) remains a foundational algorithm in machine learning, its convergence can be slow in large-scale settings due to high gradient variance (Johnson & Zhang, 2013). Addressing this, the variance-reduced (VR) methods (Gower et al., 2020), notably SVRG (Johnson & Zhang, 2013; Allen-Zhu & Yuan, 2016; Reddi et al., 2016) and SAGA (Defazio et al., 2014), represent a significant theoretical advancement. These algorithms reduce variance by leveraging gradients from past iterates or periodically computing full-batch gradients and achieve faster convergence rates (e.g., linear convergence under certain assumptions) compared to SGD (Johnson & Zhang, 2013; Reddi et al., 2016). Despite these theoretical benefits, a primary practical limitation is the substantial computational and memory cost associated with full-batch gradients. This overhead can become prohibitive as model sizes and datasets scale. Another common extension of SGD is the use of adaptive step-sizes, as in Adagrad (Duchi et al., 2011) and ADAM (Kingma & Ba, 2014). However, the convergence properties of these adaptive methods remain debated and can be highly sensitive to hyper-parameter choices (Reddi et al., 2019; Défossez et al., 2020; Zhang et al., 2022). To provide a clearer and more interpretable comparison, we focus on standard baselines such as SGD and SVRG in our theoretical analysis, which better isolate the effects of our proposed modifications.

**Zeroth-order optimization.** Zeroth-order (ZO) optimization approximates gradients using function evaluations instead of explicit gradient computation, offering reduced computational and memory overhead (Zhang et al., 2024). This advantage makes them attractive for large-scale problems like fine-tuning of large language models (Malladi et al., 2023; Gautam et al., 2024; Zhang et al., 2024; Tang et al., 2024; Ling et al., 2024), and their convergence properties are theoretically studied (Duchi et al., 2015; Jamieson et al., 2012; Nesterov & Spokoiny, 2017; Liu et al., 2018b). However, the convergence rates of ZO methods often degrade with increasing parameter dimension $d$, which makes them slow for large models compared to FO methods (Liu et al., 2018b;a; Wang et al., 2018). Even recent applications like MeZO (Malladi et al., 2023) and MeZO-SVRG (Gautam et al., 2024) are constrained by large parameter dimension. ZO optimization is crucial for black-box scenarios (Chen et al., 2017; Tu et al., 2019), while tasks like fine-tuning models often have accessible gradients whose computation is merely expensive. This motivates our hybrid approach, which aims to combine ZO's efficiency with FO's faster convergence by strategically incorporating both types of gradient information, thereby avoiding the high computational cost associated with FO methods, while also mitigating the performance degradation that ZO methods often suffer in high-dimensional settings.

**Hybrid Zeroth-Order and First-Order Algorithms.** Combining the strengths of FO and ZO optimization is a relatively recent and underexplored direction, with limited established theoretical analysis. The goal is to enjoy faster convergence while using less computational resource via ZO

techniques (Zhang et al., 2024; Li et al., 2024). Early explorations include schemes like applying ZO to shallower model layers and FO to deeper ones (Zhang et al., 2024), or concurrently computing and combining FO-SGD and ZO-SGD updates at each step, as in Addax (Li et al., 2024). However, these initial hybrid approaches may have limitations; for instance, the theoretical convergence rate of Addax still exhibits dependence on the parameter dimension $d$ (Li et al., 2024), hindering its effectiveness for large-scale models. Furthermore, many early hybrid strategies often lack explicit mechanisms to adaptively tune the balance between FO accuracy and ZO query efficiency in response to varying computational resources or specific problem demands. The scarcity of hybrid strategies that offer both theoretical convergence and controlled adaptability underscores the novelty of our work. To contextualize our contributions, Table 1 summarizes the convergence rates and computational complexities of our proposed methods, referred to as VAMO and VAMO (multi-point) in the table alongside several FO and ZO algorithms. We provide a detailed explanation of Table 1 in Appendix C.

Table 1: Summary of convergence rate and computational complexity of our proposals given $T$ total iterations. $n$ represents the total number of samples or component functions, $d$ is the parameter dimension, $b$ denotes the mini-batch size, $S$ is the number of epochs or outer loops (for SVRG-type methods, $T \approx Sm$ where $m$ is the number of inner iterations per epoch), and $q$ signifies the number of query directions used for ZO estimation.

| Method | Grad. estimator | Stepsize | Convergence rate (worst case as $b < n$) | Computational complexity |
|---|---|---|---|---|
| ZO-SVRG | Gradient Estimate | $O(1/d)$ | $O(d/T + 1/b)$ | $O(nS + 2bT)$ |
| FO-SGD | Explicit Gradient | $O(1/\sqrt{T})$ | $O(1/\sqrt{T})$ | $O(bdT)$ |
| FO-SVRG | Explicit Gradient | $O(1)$ | $O(1/T)$ | $O(dnS + 2bdT)$ |
| VAMO | Mixed Gradient | $O(1)$ | $O(1/T + 1/b)$ | $O(nS + bT + bdT)$ |
| VAMO(multi-point) | Mixed Gradient | $O(1)$ | $O(1/T + (1 - q/d)^2/b)$ | $O(qnS + bqT + bdT)$ |

## 3  PRELIMINARIES

We consider the following nonconvex finite-sum optimization problem:

$$\min_{\mathbf{x} \in \mathbb{R}^d} \quad f(\mathbf{x}) := \frac{1}{n} \sum_{i=1}^{n} f_i(\mathbf{x}), \tag{1}$$

where $\{f_i(\mathbf{x})\}_{i=1}^{n}$ are $n$ individual nonconvex cost functions. Note that equation 1 is the generic form of many machine learning problems such as training neural networks, since this is the natural form arising from empirical risk minimization (ERM). Next we introduce assumptions we will make throughout the paper and provide the background of ZO gradient estimate.

### 3.1  ASSUMPTIONS

Throughout this paper, we make the following standard assumptions on the objective function components $f_i(\mathbf{x})$. Let $d$ be the dimension of the optimization variable $\mathbf{x}$.

**Assumption 1** (L-smooth). *Each function $f_i : \mathbb{R}^d \to \mathbb{R}$ is L-smooth for $i \in [n] := \{1, 2, \ldots, n\}$. That is, for any $\mathbf{x}, \mathbf{y} \in \mathbb{R}^d$, there exists a constant $L > 0$ such that:*

$$\|\nabla f_i(\mathbf{x}) - \nabla f_i(\mathbf{y})\|_2 \le L\|\mathbf{x} - \mathbf{y}\|_2$$

*This also implies that the full objective function $f(\mathbf{x}) = \frac{1}{n} \sum_{i=1}^{n} f_i(\mathbf{x})$ is L-smooth.*

**Assumption 2** (Bounded Variance). *The variance of the stochastic gradients is bounded. Specifically, for any $\mathbf{x} \in \mathbb{R}^d$, there exists a constant $\sigma^2 \ge 0$ such that:*

$$\frac{1}{n} \sum_{i=1}^{n} \|\nabla f_i(\mathbf{x}) - \nabla f(\mathbf{x})\|_2^2 \le \sigma^2$$

*Here, $\nabla f_i(\mathbf{x})$ is the gradient of a single component function, which can be viewed as a stochastic gradient of $f(\mathbf{x})$ if $i$ is chosen uniformly at random from $[n]$.*

These assumptions are standard in the analysis of stochastic optimization algorithms for nonconvex problems (Ghadimi & Lan, 2013; Reddi et al., 2016; Nesterov & Spokoiny, 2017).

## 3.2 Convergence Notion

We study the nonconvex optimization problem in equation 1, which is common in modern machine learning. For convex problems, convergence is typically measured by the expected suboptimality $\mathbb{E}[f(\mathbf{x}_T) - f(\mathbf{x}^*)]$. In contrast, nonconvex problems often contain multiple local minima and saddle points (Dauphin et al., 2014; Balasubramanian & Ghadimi, 2022), making global optimality intractable. Consequently, convergence is evaluated by the first-order stationarity condition via the expected squared gradient norm $\mathbb{E}[\|\nabla f(\mathbf{x})\|_2^2]$. An algorithm is deemed convergent when this metric approaches zero or falls below a tolerance $\epsilon$ (Liu et al., 2020; Mu et al., 2024), which is the criterion used in our theoretical guarantees.

## 3.3 ZO Gradient Estimation

Consider an individual cost function $f_i : \mathbb{R}^d \to \mathbb{R}$ that satisfies the conditions in Assumption 1. The ZO approach estimates gradients using only function evaluations. A commonly used two-point ZO gradient estimator for $f_i(\mathbf{x})$ is defined as (Spall, 1992; Nesterov & Spokoiny, 2017):

$$\hat{\nabla} f_i(\mathbf{x}) = \frac{d}{\mu} \left[ f_i(\mathbf{x} + \mu \mathbf{u}_i) - f_i(\mathbf{x}) \right] \mathbf{u}_i, \quad \text{for } i \in [n], \tag{2}$$

where $d$ is the dimension of the optimization variable $\mathbf{x}$, $\mu > 0$ is a small smoothing parameter, and $\{\mathbf{u}_i\}_{i=1}^n$ are i.i.d. random direction vectors drawn uniformly from the unit Euclidean sphere in $\mathbb{R}^d$ (i.e., $\mathbf{u}_i \sim U(\mathbb{S}^{d-1})$) (Flaxman et al., 2004; Shamir, 2017; Gao et al., 2018).

In general, for $\mu > 0$, the ZO gradient estimator $\hat{\nabla} f_i(\mathbf{x})$ is a biased approximation of the true gradient $\nabla f_i(\mathbf{x})$. The bias tends to decrease as $\mu \to 0$. However, in practical implementations, choosing an excessively small $\mu$ can render the function difference $f_i(\mathbf{x} + \mu \mathbf{u}_i) - f_i(\mathbf{x})$ highly sensitive to numerical errors or system noise or numerical precision issues, potentially failing to accurately represent the local change in the function (Lian et al., 2016). A key property of the ZO estimator is that for $\mu > 0$, it provides an unbiased estimate of the gradient of a smoothed version of $f_i$, often denoted $f_{i,\mu}(\mathbf{x}) = \mathbb{E}_{\mathbf{v}}[f_i(\mathbf{x} + \mu \mathbf{v})]$ (where $\mathbf{v}$ is a random vector from a unit ball or sphere), i.e., $\mathbb{E}_{\mathbf{u}_i}[\hat{\nabla} f_i(\mathbf{x})] = \nabla f_{i,\mu}(\mathbf{x})$ (Liu et al., 2020).

To reduce the error of the ZO gradient estimate, a multi-point version can be employed. Instead of using a single random direction $\mathbf{u}_i$, $q \geq 1$ i.i.d. random directions $\{\mathbf{u}_{i,j}\}_{j=1}^q$ are sampled for each $f_i$. Since estimating along each direction requires two function queries, the multi-point ZO gradient estimator involves a total of $2q$ function queries and is defined as (Duchi et al., 2015; Liu et al., 2020):

$$\hat{\nabla} f_i(\mathbf{x}) = \frac{d}{\mu q} \sum_{j=1}^q \left[ f_i(\mathbf{x} + \mu \mathbf{u}_{i,j}) - f_i(\mathbf{x}) \right] \mathbf{u}_{i,j}, \quad \text{for } i \in [n]. \tag{3}$$

We refer to this as the multi-point ZO gradient estimate throughout the paper.

## 3.4 Notations

In this paper, we denote $\nabla f(x), \nabla f_i(x)$ as FO gradients of $f(x)$ and $f_i(x)$, respectively and $\hat{\nabla} f(x), \hat{\nabla} f_i(x)$ as their ZO variants. $\mathbb{E}[\cdot]$ operates as the usual mathematical expectation, and $\mathcal{I}$ is a mini-batch of indices sampled from $[n] := 1, \ldots, n$, with size $b = |\mathcal{I}|$. $\| \cdot \|_2$ denotes the Euclidean (i.e., $\ell_2$) norm.

## 4 Hybrid FO and ZO Stochastic Variance Reduction (VAMO)

### 4.1 From SVRG and ZO-SVRG to Hybrid SVRG

The principles of FO-SVRG and ZO-SVRG have been extensively explored in optimization literature (Johnson & Zhang, 2013; Reddi et al., 2016; Liu et al., 2018b; Ji et al., 2019). FO-SVRG, in

particular, is known to achieve a linear convergence rate $O(1/T)$ for non-convex problems under certain conditions, significantly outperforming the convergence rate of FO-SGD (Reddi et al., 2016). The key step of FO-SVRG involves leveraging a full gradient $\nabla f(\hat{\mathbf{x}})$, computed periodically at the snapshot $\hat{\mathbf{x}}$, to construct a variance-reduced stochastic gradient estimate (Johnson & Zhang, 2013):

$$\hat{\mathbf{g}}_{\text{FO-SVRG}} = \nabla f_{\mathcal{I}}(\mathbf{x}) - \nabla f_{\mathcal{I}}(\hat{\mathbf{x}}) + \nabla f(\hat{\mathbf{x}}), \tag{4}$$

where $\nabla f_{\mathcal{I}}(\mathbf{x}) = \frac{1}{b}\sum_{i\in\mathcal{I}}\nabla f_i(\mathbf{x})$ is the mini-batch stochastic gradient from a subset $\mathcal{I} \subseteq [n]$ of size $b$. A crucial property of $\hat{\mathbf{g}}_{\text{FO-SVRG}}$ is that $\hat{\mathbf{g}}_{\text{FO-SVRG}}$ is an unbiased gradient estimate of $\nabla f(\mathbf{x})$, $\mathbb{E}[\hat{\mathbf{g}}_{\text{FO-SVRG}}] = \nabla f(\mathbf{x})$ (Johnson & Zhang, 2013).

In the ZO setting, ZO-SVRG adapts the SVRG structure by replacing all explicit gradient computations with ZO estimates derived from function evaluations:

$$\hat{\mathbf{g}}_{\text{ZO-SVRG}} = \hat{\nabla} f_{\mathcal{I}}(\mathbf{x}) - \hat{\nabla} f_{\mathcal{I}}(\hat{\mathbf{x}}) + \hat{\nabla} f(\hat{\mathbf{x}}), \tag{5}$$

where $\hat{\nabla} f_{\mathcal{I}}(\mathbf{x}) = (1/b)\sum_{i\in\mathcal{I}}\hat{\nabla} f_i(\mathbf{x})$, $\hat{\nabla} f(\mathbf{x}) = \hat{\nabla} f_{[n]}(\mathbf{x})$, and $\hat{\nabla} f_i(\mathbf{x})$ is a ZO gradient estimate, as defined in Section 3.3. While structurally similar, a key distinction is that $\hat{\mathbf{g}}_{\text{ZO-SVRG}}$ is generally a biased estimate of $\nabla f(\mathbf{x})$ due to the inherent bias of $\hat{\nabla} f_i(\mathbf{x})$ relative to $\nabla f_i(\mathbf{x})$ (Liu et al., 2020). This bias significantly complicates its convergence analysis compared to FO-SVRG.

VAMO (Algorithm 1) is motivated by the high cost of computing the full gradient $\nabla f(\hat{\mathbf{x}})$ in SVRG for large-scale models, and introduces a hybrid gradient estimator that combines FO and ZO components to reduce this overhead:

$$\hat{\mathbf{g}} = \nabla f_{\mathcal{I}}(\mathbf{x}) - \alpha\left(\hat{\nabla} f_{\mathcal{I}}(\hat{\mathbf{x}}) - \hat{\nabla} f(\hat{\mathbf{x}})\right). \tag{6}$$

Here, $\nabla f_{\mathcal{I}}(\mathbf{x})$ is the standard FO mini-batch stochastic gradient at the current iterate $\mathbf{x}$, while $\hat{\nabla} f(\hat{\mathbf{x}})$ is the ZO estimate of the full gradient at the snapshot $\hat{\mathbf{x}}$.

A critical design choice in equation 6 is the construction of the variance-reduction term $\alpha\left(\hat{\nabla} f_{\mathcal{I}}(\hat{\mathbf{x}}) - \hat{\nabla} f(\hat{\mathbf{x}})\right)$. To preserve the desirable unbiased property of FO-SVRG, we ensure that the expectation of the ZO variance correction term, $\mathbb{E}\left[\hat{\nabla} f_{\mathcal{I}}(\hat{\mathbf{x}}) - \hat{\nabla} f(\hat{\mathbf{x}})\right]$, is zero. Using ZO estimates for both terms within the parentheses, $\hat{\nabla} f_{\mathcal{I}}(\hat{\mathbf{x}})$ and $\hat{\nabla} f(\hat{\mathbf{x}})$, is key to this property and also contributes to computational savings at the snapshot. Maintaining this unbiasedness is pivotal, as it allows for a more tractable convergence analysis similar to FO-SVRG, distinguishing our approach from many ZO algorithms that contend with biased estimators. Furthermore, VAMO introduces a novel mixing coefficient $\alpha > 0$. This parameter allows for explicit control over the influence of the ZO variance correction term. We will provide a detailed theoretical and empirical analysis of $\alpha$ in Appendix K.3.

*The introduction of this hybrid structure, particularly the ZO estimation at snapshot and the mixing coefficient $\alpha$, means that the convergence analysis of VAMO cannot be trivially inherited from existing FO-SVRG or ZO-SVRG analyses.* It requires a dedicated theoretical investigation to characterize its behavior and prove its convergence guarantees, which constitutes a core part of our contribution in Section 4.2. This distinct analytical challenge underscores the theoretical novelty of our work.

---

**Algorithm 1** VAMO $(T, m, \{\eta_k\}, b, \bar{\mathbf{x}}_0, \mu, \alpha)$

---

1: **Input:** In addition to parameters in SVRG, set smoothing parameter $\mu > 0$.
2: **for** $s = 1, 2, \ldots, S$ **do**
3:     compute ZO estimate $\hat{\mathbf{g}}_s = \hat{\nabla} f(\bar{\mathbf{x}}_{s-1})$
4:     set $\mathbf{x}_0^s = \bar{\mathbf{x}}_{s-1}$
5:     **for** $k = 0, 1, \ldots, m-1$ **do**
6:         choose mini-batch $I_k$ of size $b$
7:         compute hybrid FO and ZO gradient blending: $\mathbf{v}_k^s = \nabla f_{I_k}(\mathbf{x}_k^s) - \alpha(\hat{\nabla} f_{I_k}(\mathbf{x}_0^s) - \hat{\mathbf{g}}_s)$
8:         update $\mathbf{x}_{k+1}^s = \mathbf{x}_k^s - \eta_k \mathbf{v}_k^s$
9:     **end for**
10:    set $\bar{\mathbf{x}}_s = \mathbf{x}_m^s$
11: **end for**
12: return $\bar{\mathbf{x}}$ chosen uniformly at random from $\{\{\mathbf{x}_k^s\}_{k=0}^{m-1}\}_{s=1}^S$.

---

## 4.2 Convergence analysis

In this section, we present the convergence analysis for VAMO using the two-point ZO gradient estimate in equation 2. Our analysis is based on an upper bound on the expected squared gradient norm $\mathbb{E}\big[\|\nabla f(\bar{\mathbf{x}})\|_2^2\big]$, as shown in Theorem 1. As discussed in Section 3.2, for non-convex objectives, a small value of $\mathbb{E}\big[\|\nabla f(\bar{\mathbf{x}})\|_2^2\big]$ implies convergence to a stationary point.

**Theorem 1.** *Under the assumptions in Section 3.1, and the two-point ZO gradient estimate is used. The output $\bar{\mathbf{x}}$ of Algorithm 1 satisfies:*

$$\mathbb{E}\big[\|\nabla f(\bar{\mathbf{x}})\|_2^2\big] \leq \frac{\mathbb{E}[f(\bar{\mathbf{x}}_0) - f^*]}{T\bar{\gamma}} + \frac{S\chi_m}{T\bar{\gamma}}, \tag{7}$$

*where $T = Sm$, $f^* = \min_{\mathbf{x}} f(\mathbf{x})$, $\bar{\gamma} = \min_{k \in [m]} \gamma_k$, and $\chi_m = \sum_{k=0}^{m-1} \chi_k$. $\gamma_k$ and $\chi_k$ are coefficients which depend on $\{\eta_k, \mu, b, d, \alpha\}$. The proof is provided in Appendix F.*

Compared to FO-SVRG, Theorem 1 has an additional error $(S\chi_m/(T\bar{\gamma}))$ due to the use of the ZO gradient estimator. To obtain a clear dependence on these parameters and explore deeper insights into convergence, we simplify equation 7 to suit the specific parameter settings, as shown below.

**Corollary 1.** *Suppose parameters are set as*

$$\mu = \frac{1}{\sqrt{T}}, \quad \eta_k = \eta = \frac{\rho}{L}, \quad \alpha = \frac{1}{d}, \tag{8}$$

*with $\beta_k = \beta = L$, where $0 < \rho \leq 1$ is a universal constant independent of $b$, $d$, $\alpha$, $L$ and $T$. Then Theorem 1 implies*

$$\frac{\mathbb{E}[f(\bar{\mathbf{x}}_0) - f^*]}{T\bar{\gamma}} \leq O\left(\frac{1}{T}\right), \quad \frac{S\chi_m}{T\bar{\gamma}} \leq O\left(\frac{1}{bT} + \frac{1}{b}\right), \tag{9}$$

*yielding the convergence rate:*

$$\mathbb{E}\big[\|\nabla f(\bar{\mathbf{x}})\|_2^2\big] \leq O\left(\frac{1}{T} + \frac{1}{bT} + \frac{1}{b}\right). \tag{10}$$

*The proof is provided in Appendix G.*

From Corollary 1, we can observe that one advantage of VAMO is that, compared to previous ZO algorithms, the value of smoothing parameters $\mu$ is less restrictive. For example, ZO-SVRG required $\mu \leq O(1/\sqrt{dT})$, and ZO-SGD required $\mu \leq O(1/d\sqrt{T})$ (Liu et al., 2018b). Compared to FO-SGD, the algorithm achieves an improved rate of $O(1/T)$ rather than the rate of $O(1/\sqrt{T})$. Compared to ZO algorithms, the convergence rate is independent of the parameter dimension $d$. Compared to FO-SVRG, VAMO suffers an additional error of $O(1/b)$ inherited from $(S\chi_m/(T\bar{\gamma}))$ in equation 1.

## 4.3 Memory Efficiency of VAMO

Building on the convergence analysis of VAMO in the Section 4.2, where we showed that VAMO achieves an $O(1/T)$ convergence rate, we now analyze its memory efficiency in large-scale training.

In large-scale training, the total memory footprint of an algorithm is dominated by three components: model parameters $|\mathbf{x}|$, optimizer states, and dynamic memory for gradient computation (Zhang et al., 2024). Here $|\mathbf{x}|$ denotes the memory required to store the model parameters in full precision, and $|\mathbf{x}_l|$ is that of the parameters in layer $l$. The symbol $|\mathbf{a}_l|$ denotes the memory of activations or intermediate results of one single sample at layer $l$.

For FO methods, dynamic memory is dominated by intermediate results, scaling roughly as $\sum_l b \cdot |\mathbf{a}_l|$ in Table 1 for a mini-batch of size $b$. FO-SVRG (Johnson & Zhang, 2013) further requires computing the full gradient at the snapshot point; while this can be accumulated over smaller mini-batches to reduce peak memory of intermediate results to $\sum_l b \cdot |\mathbf{a}_l|$, it increases computational cost per epoch due to multiple passes over all $n$ samples. If the full FO gradient is computed at once, the dynamic memory of intermediate results will increase to $\sum_l n \cdot |\mathbf{a}_l|$, which poses a substantial memory burden for large-scale settings. Therefore, FO-SVRG must trade off compute and memory when computing

full FO gradients. In contrast, VAMO leverages full ZO gradient estimates, which do not require backpropagation, and thus do not need to store intermediate results. Even for estimation of full batch gradients at once, dynamic memory only reaches $\max_l |\mathbf{x}_l|$, independent of $b$, the same as ZO-SGD, which is much smaller than $\sum_l b \cdot |\mathbf{a}_l|$. Therefore, VAMO do not need to introduce a compute–memory trade-off like FO-SVRG in large-scale settings. Regarding optimizer states, VAMO needs $2|\mathbf{x}|$ for storing full ZO gradient and the snapshot point, while Adagrad and Adam which need around $2|\mathbf{x}|$ to store first- or second-momentum. However, in practical applications, for faster computation, Adam and Adagrad usually need to allocate large temporary buffers after activations have already filled and fragmented GPU memory, spiking the true peak memory . In Table 2, we can observe that Adam and Adagrad bring much higher peak memory cost in model fine-tuning experiments. And Table 4 summarizes the three memory components for different optimizers and a more detailed theoretical and empirical analysis of memory efficiency is provided in Appendix B.

## 5  VAMO with Multi-Point ZO Gradient Estimation

Building upon the VAMO algorithm previously introduced with a two-point ZO gradient estimator, this section presents its multi-point ZO estimation variant. This extension is a key component of VAMO's **adaptive** design, as adjusting the number of random directions $q$ in ZO estimate allows for explicit tuning of the trade-off between computational cost and the precision of the ZO-based variance reduction, thereby directly influencing convergence performance.

**Theorem 2.** *Suppose assumptions A1 and A2 hold, and the multi-point ZO gradient estimate is used in Algorithm 1. The gradient norm bound in equation 7 yields the simplified convergence rate:*

$$\mathbb{E}\big[\|\nabla f(\bar{\mathbf{x}})\|_2^2\big] \leq O\bigg(\frac{1}{T} + \frac{1}{bT} + \frac{1}{b}\Big(1 - \frac{q}{d}\Big)^2\bigg). \tag{11}$$

*With parameter choices $\mu = \frac{1}{q\sqrt{T}}$, $\eta = \frac{\rho}{L}$, $\alpha = \frac{q}{d}$. The proof is provided in Appendix H.*

By contrast with Corollary 1, it can be seen from equation 11 that the use of multi-point version of VAMO reduces the error $O(1/b)$ in equation 1 by leveraging multiple $q$ direction sampling, while increasing the computational cost accordingly. If $q = d$, the algorithm's convergence rate become comparable to FO-SVRG. A comprehensive summary and comparison of the computational complexities and convergence rates of our proposed VAMO methods against various FO and ZO algorithms can be found in Table 1 presented in Section 2. Briefly, the mixing coefficient $\alpha$ governs the trade-off between the FO stochastic gradient and the ZO variance-correction: larger $\alpha$ increases reliance on ZO information while smaller $\alpha$ favors FO updates to mitigate the error of ZO estimation. A detailed theoretical and empirical analysis of $\alpha$ is provided in Appendix K.3.

## 6  Applications and Experiments

In this section, we present empirical results to validate the effectiveness and adaptability of the proposed VAMO through three experiments[1]: (i) an adaptability study on a synthetic task varying the number of ZO query directions $q$, (ii) a benchmark on MNIST classification (LeCun et al., 1998), and (iii) large-scale fine-tuning of GPT-2, GPT-2 Medium (Radford et al., 2019) and RoBERTa (Liu et al., 2019) on SST-2 (Socher et al., 2013) and MNLI (Williams et al., 2017). We benchmark against popular FO methods (FO-SGD (Robbins & Monro, 1951), FO-Adagrad (Duchi et al., 2011), FO-Adam (Kingma & Ba, 2014)) and ZO methods (ZO-SGD (Ghadimi & Lan, 2013), ZO-SVRG (Liu et al., 2018b)). Full setups and hyperparameters are deferred to Appendix J.

**Adaptability Experiment.**   To highlight VAMO's adaptive nature, we tested its multi-point ZO estimation strategy on a synthetic non-convex least-squares task. We compared variants using $q \in \{1, 3, 5\}$ query directions against FO-SGD. Fig. 1a presents the training loss convergence. Consistent with our theoretical analysis in Table 1, all VAMO variants achieve an $O(1/T)$ convergence rate, outperforming FO-SGD's $O(1/\sqrt{T})$ rate. The figure clearly illustrates VAMO's adaptability: increasing $q$ improves convergence performance, effectively mitigating the additional $O(1/b)$ error

---

[1]Code for all experiments is available in the supplementary material.

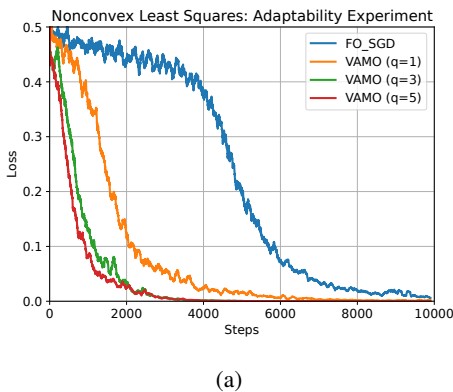 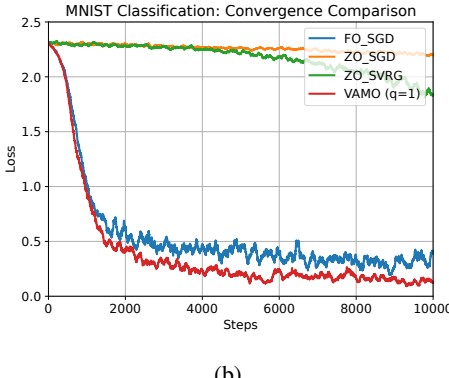

(a)                                                    (b)

Figure 1: (a) Convergence comparison on a non-convex least-squares task, showing VAMO with varying ZO query points ($q = 1, 3, 5$) against FO-SGD. (b) Convergence comparison on the MNIST classification task against pure FO and ZO methods.

term associated with the two-point ($q = 1$) variant. This aligns with the theoretical prediction that this error term diminishes for larger $q$ (scaling towards $O((1 - q/d)^2/b)$). These results empirically validate our theory and highlight VAMO's practical ability to adaptively trade computational cost for enhanced convergence by managing ZO estimation error, a key aspect of its adaptive design.

**Multiclass Classification.** On the MNIST benchmark, we trained a Multi-Layer Perceptron (MLP) and compared VAMO (two-point version, $q = 1$) against FO optimizer (FO-SGD) and ZO optimizers (ZO-SGD and ZO-SVRG). As shown in Fig. 1b, VAMO significantly outperforms purely ZO methods, converging faster and reaching a lower final loss. Its performance is also highly competitive with FO-SGD, underscoring the practical effectiveness of our hybrid strategy.

Table 2: We measure peak GPU memory consumption (in GB) when fine-tuning RoBERTa-Large and GPT-2 Medium models under varying batch sizes (bs) with a fixed context length (cl=128).

| Method | RoBERTa-Large | | | GPT-2-Medium | | |
|---|---|---|---|---|---|---|
| | bs = 16 | bs = 32 | bs = 64 | bs = 16 | bs = 32 | bs = 64 |
| FO-SGD | 4.63 | 6.72 | 10.83 | 7.08 | 11.50 | 20.33 |
| FO-Adagrad | 7.21 | 10.46 | 16.54 | 10.25 | 15.74 | 27.57 |
| FO-Adam | 8.62 | 11.59 | 17.66 | 11.59 | 17.05 | 29.07 |
| **VAMO** | **5.95** | **7.83** | **11.96** | **8.40** | **12.62** | **21.46** |

Table 3: We show test accuracies and losses (in parentheses) of VAMO when fine-tuning GPT-2, GPT-2 Medium and RoBERTa-Large on SST-2 and MNLI datasets, compared with FO and ZO baselines.

| Method | RoBERTa-Large (MNLI) | GPT-2 Medium (MNLI) | GPT-2 (SST-2) |
|---|---|---|---|
| FO-SGD | 75 (0.67) | 60 (0.90) | 90 (0.42) |
| FO-Adagrad | 72 (0.74) | 53 (1.33) | 88 (0.37) |
| FO-Adam | 61 (0.98) | 48 (1.03) | 98 (0.03) |
| ZO-SGD | 73 (0.99) | 66 (0.89) | 84 (0.59) |
| ZO-SVRG | 70 (1.01) | 69 (0.88) | 72 (0.64) |
| VAMO | 78 (0.56) | 67 (0.85) | 94 (0.16) |

**Fine-Tuning Experiments.** To further assess VAMO's practical utility and its advantages in complex, large-scale settings, we further evaluate VAMO on fine-tuning pre-trained large models, including GPT-2, GPT-2 Medium, and RoBERTa-Large, on the SST-2 and MNLI datasets, comparing

against standard FO optimizers (FO-SGD, FO-Adagrad, FO-Adam) and ZO optimizers (ZO-SGD, ZO-SVRG). Table 2 presents the peak GPU memory usage of different algorithms for batch sizes of 16, 32, and 64, with the context length fixed at 128. Compared with FO-Adam and FO-Adagrad, VAMO showcase great memory efficiency, The results show that VAMO's peak memory cost is 32% (RoBERTa-Large) and 26% (GPT-2 Medium) lower than FO-Adam on average. Compared to FO-Adagrad, the memory cost of VAMO is 23% (RoBERTa-Large) and 20% (GPT-2 Medium) lower on average.

Fig. 2 presents the training loss of different algorithms over both iteration steps and wall-clock time. VAMO achieves markedly faster and more stable convergence than FO-SGD, while maintaining a similar memory footprint. This improvement is theoretically supported by Table 1: VAMO enjoys a convergence rate of $\mathcal{O}(1/T)$, superior to the $\mathcal{O}(1/\sqrt{T})$ rate of FO-SGD. Compared with ZO methods, VAMO not only yields better empirical performance but also avoids dependence on the parameter dimension $d$, demonstrating stronger scalability for large models. Compared with Adagrad, VAMO achieves a similar level of performance but with far lower memory overhead.

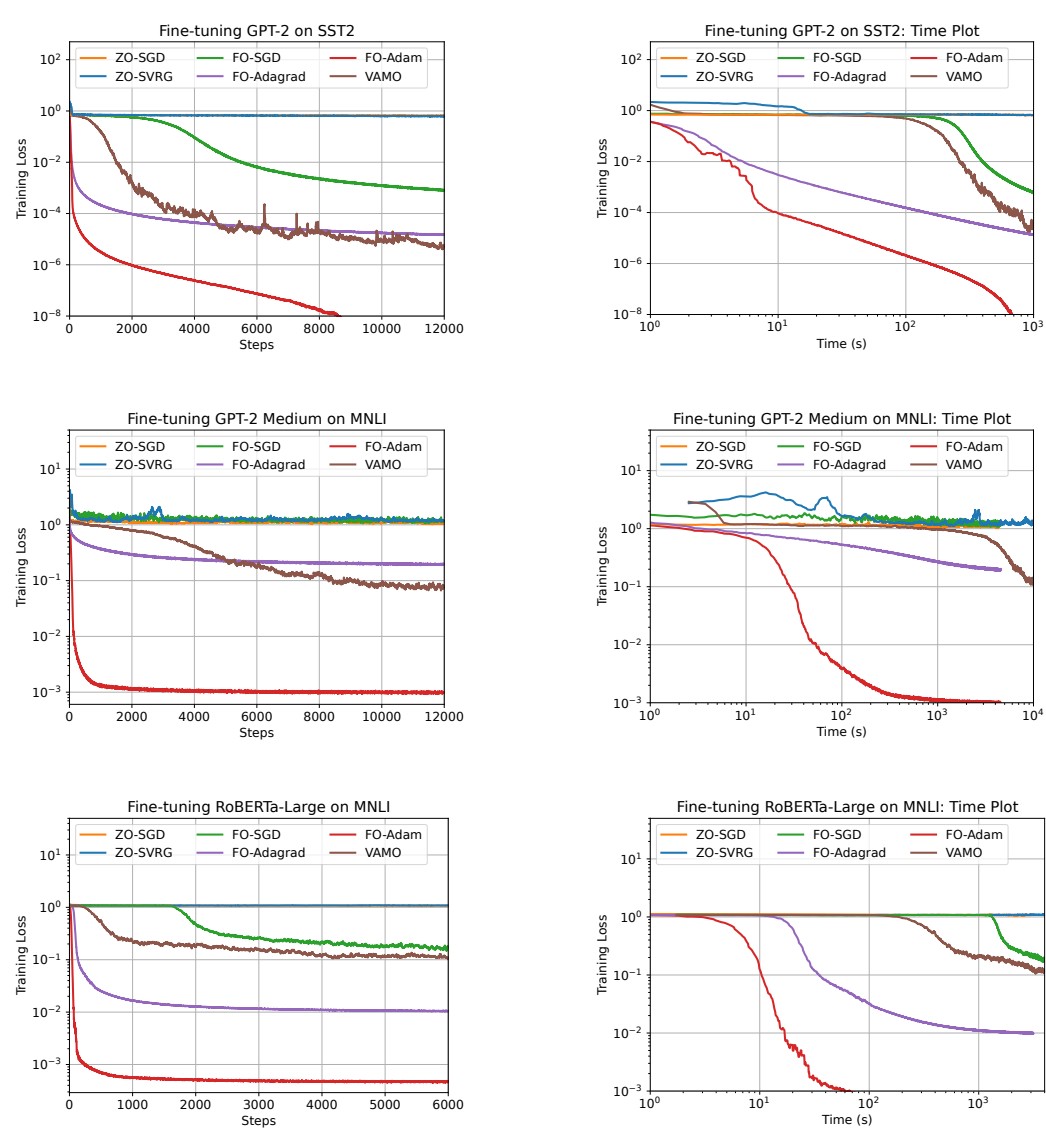

Figure 2: We compare the convergence performance of VAMO when fine-tuning GPT-2, GPT-2 Medium and RoBERTa-Large on SST-2 and MNLI datasets with pure FO and ZO methods.

Table 3 shows the final test accuracies and test losses (in parentheses) of different algorithms for all tasks. Although VAMO underperforms FO-Adam in convergence speed, it demonstrates better generalization ability in most tasks. For instance, VAMO's test accuracy is $8\%$ higher than that of FO-Adam in RoBERTa-Large experiments and $26\%$ higher in the GPT-2 Medium (MNLI) experiments, while also achieving lower test losses in both. Compared with other algorithms, VAMO not only outperforms them in wall-clock convergence, it also achieves a near $10\%$ higher accuracy score on average, demonstrating both superior convergence speed and better generalization with a light memory footprint. We believe that the improved generalization ability might come from the use of full-batch gradients, since all other algorithms only use mini batch gradients, which may result in insufficient training.

## 7    CONCLUSION

In this paper, we propose a hybrid FO and ZO variance-reduced algorithm, VAMO, for nonconvex optimization. We demonstrate that compared to FO-SGD, our algorithm improves the convergence rate from $O(1/\sqrt{T})$ to a linear rate of $O(1/T)$. Compared to ZO algorithms, our method maintains convergence performance independent of the parameter dimension $d$, making it effective for optimizing high-dimensional problems. However, due to the use of two-point ZO gradient estimation, our convergence result includes an additional error term $O(1/b)$. To mitigate this, we introduce a multi-point ZO gradient estimation variant, which reduces this error. Unlike previous purely FO or ZO methods, our hybrid approach leverages the advantages of both, enabling a more flexible trade-off between efficiency and convergence performance. This makes it more adaptable to real-world applications with complex constraints. Our theoretical analysis and empirical evaluations, including comparisons with state-of-the-art methods, demonstrate the effectiveness of our approach.

## REPRODUCIBILITY STATEMENT

We have provided detailed descriptions of our algorithms, training settings, and datasets in the main text and appendix. In addition, all source code and instructions to reproduce our experiments are included in the supplementary material.

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

## A    THE USE OF LARGE LANGUAGE MODELS

The use of large language models (LLMs) in this work was limited to writing assistance and minor code editing. Theoretical analysis and experimental results were conducted without LLM involvement.

## B    DETAILED DISCUSSION ON MEMORY EFFICIENCY

We have made a brief analysis of VAMO's memory efficiency in Section 4.3. In this section, we analyse VAMO's memory profile in detail and compare it against standard FO and ZO optimizers, theoretically and empirically.

### B.1    THEORETICAL ANALYSIS

We adapt the memory analysis framework of Zhang et al. (2024) to compare the peak memory consumption of different optimizers. Table 4 summarizes the decomposition into three components: the model (Weight Mem.), the optimizer states (Opt. State Mem.), and dynamic allocation for computing gradients and optimization (Dynamic Mem.).

Table 4: Comparison of the instant peak memory consumption of different optimizers when fine-tuning the full model. Here $|\mathbf{x}|$ denotes the memory required to store the model parameters in full precision, and $|\mathbf{x}_l|$ is that of the parameters in layer $l$. The symbol $|\mathbf{a}_l|$ denotes the memory of intermediate activations of one single sample at layer $l$, where $b$ and $n$ correspond to the mini-batch size and the total dataset size, respectively. In Dynamic Mem., $b \cdot |\mathbf{a}_l|$ represents the memory of saving mini-batch activations at layer $l$, while $|\mathbf{x}_l|$ also accounts for temporarily saved gradients, as they have the same size as the corresponding parameters (Zhang et al., 2024).

| Optimizer | Weight Mem. | Dynamic Mem. (Grad.&Opt.) | Opt. State Mem. |
|---|---|---|---|
| FO-SGD | $|\mathbf{x}|$ | $\sum_l \max\{b \cdot |\mathbf{a}_l|, |\mathbf{x}_l|\}$ | 0 |
| FO-SVRG (accumulation) | $|\mathbf{x}|$ | $\sum_l \max\{b \cdot |\mathbf{a}_l|, |\mathbf{x}_l|\}$ | $2|\mathbf{x}|$ |
| FO-SVRG (full batch at once) | $|\mathbf{x}|$ | $\sum_l \max\{n \cdot |\mathbf{a}_l|, |\mathbf{x}_l|\}$ | $2|\mathbf{x}|$ |
| FO-Adam | $|\mathbf{x}|$ | $\sum_l \max\{b \cdot |\mathbf{a}_l|, |\mathbf{x}_l|\}$ | $2|\mathbf{x}|$ |
| FO-Adagrad | $|\mathbf{x}|$ | $\sum_l \max\{b \cdot |\mathbf{a}_l|, |\mathbf{x}_l|\}$ | $|\mathbf{x}|$ |
| VAMO | $|\mathbf{x}|$ | $\sum_l \max\{b \cdot |\mathbf{a}_l|, |\mathbf{x}_l|\}$ | $2|\mathbf{x}|$ |
| ZO-SGD | $|\mathbf{x}|$ | $\max_l |\mathbf{x}_l|$ | 0 |

The advantage of ZO methods is the removal of backpropagation, which eliminates the need to store intermediate activations $|\mathbf{a}_l|$ for each input example during the forward pass. In FO optimizers, the total activation memory grows as $b \cdot \sum_l |\mathbf{a}_l|$ for a mini-batch of size $b$, while ZO methods completely avoid this dependence on the batch size. This reduction in dynamic memory cost becomes especially significant when $b$ is large in large-scale settings. Moreover, ZO gradients can be estimated in a layer-wise manner, so only the gradients of the currently processed layer need to be stored, further lowering the peak memory footprint compared to FO methods (Zhang et al., 2024). For ZO-SGD, dynamic memory is dominated by a temporary copy of the parameters of a single layer for gradient estimation, resulting in a peak memory of $\max_l |\mathbf{x}_l|$, independent of batch size, and no extra optimizer state is required.

In contrast, all FO methods require storing intermediate activations or other temporary variables to compute gradients, which results in dynamic memory that grows with the mini-batch size. FO-SVRG (Johnson & Zhang, 2013) additionally needs to compute the full gradient at the snapshot point. Although this full gradient can be obtained by accumulating gradients over smaller mini-batches, reducing dynamic memory of intermediate results per step to $b \cdot |\mathbf{a}_l|$, it comes at the expense of higher computational cost per epoch due to multiple passes over all mini-batches. Computing the full gradient in a single full batch would substantially increase dynamic memory of intermediate results to $n \cdot |\mathbf{a}_l|$, requiring a compute-memory trade-off as commonly done in FO-SVRG. In contrast, VAMO leverages ZO estimation to compute the full gradient without storing intermediate activations, so even when computing over the full batch at once, the peak dynamic memory is only $\max_l |\mathbf{x}_l|$,

the same as ZO-SGD, and no compute–memory trade-off is required. This makes VAMO highly memory-efficient for large mini-batches or large models. Regarding optimizer state, adaptive methods such as Adam (Kingma & Ba, 2014) or Adagrad (Duchi et al., 2011) require additional memory for first- or second-order momentum, resulting in $|\mathbf{x}|$–$2|\mathbf{x}|$ extra memory per step. FO-SVRG and VAMO need $2|\mathbf{x}|$ optimizer state for storing the snapshot and full gradient (Johnson & Zhang, 2013).

### B.2 Empirical Results Analysis

To empirically validate our theoretical analysis, we conducted **full-parameter fine-tuning** of a pre-trained RoBERTa-large model (Liu et al., 2019) and GPT-2 medium model (Radford et al., 2019) on the MNLI dataset (Williams et al., 2017) with FP32 (see Section 6). We measured the peak GPU memory consumption of VAMO, FO-SGD, FO-Adagrad and FO-Adam by varying the batch size (with sequence length fixed at 128). The detailed results are presented in Table 2 in Section 6.

We observe that VAMO's peak memory consumption is consistently only marginally higher than FO-SGD, with the difference stemming from the overhead of storing the ZO full gradient and the snapshot point for variance reduction. Importantly, as the batch size increases, the scaling trend of VAMO almost parallels that of FO-SGD, confirming that the intermediate results for computing mini-batch FO gradients dominates the memory cost in both methods. Notably, the ZO full gradient in VAMO does not require storing additional intermediate activations, further preventing the memory blow-up typically associated with FO-SVRG.

Compared with adaptive methods like Adam and Adagrad, it seems that there is no improvement of memory cost in theoretical analysis. However, in practice, VAMO achieves a substantial improvement in peak memory compared to FO-Adam and Adagrad. Adaptive methods like Adam and Adagrad, for faster computation, allocate large, temporary buffers after activations have already filled and fragmented GPU memory, spiking the true peak memory. This "last-minute" allocation often fails to reuse memory efficiently, spiking the true peak memory. VAMO's update is linear, in-place, and relies on static states. It never requests these problematic temporary buffers, thus avoiding the allocation spike entirely and resulting in a lower practical footprint. Taken together, these results highlight VAMO's practical advantage: for a negligible increase in memory compared to FO-SGD, it achieves markedly better convergence, as shown in Section 6, while remaining far more memory efficient than Adam and Adagrad. This balance makes VAMO particularly well suited for memory-constrained large-scale fine-tuning tasks.

## C Detailed Discussing of Convergence and Complexity

In this section, we provide additional discussion of the convergence rates and computational complexities summarized in Table 1 (see Section 2). Table 1 summarizes the convergence rates and computational complexities of our proposed methods, referred to as VAMO and VAMO (multi-point) in the table alongside several FO and ZO algorithms.

For ZO methods, ZO-SVRG is listed with a complexity of $O(nS + 2bT)$ function queries. Among FO methods, FO-SGD has the lowest computational cost $O(bdT)$ but also exhibits the slowest convergence rate of $O(1/\sqrt{T})$. FO-SVRG improves convergence to $O(1/T)$ but increases the cost to $O(dnS + 2bdT)$ due to full gradient computations. Our proposed VAMO maintains a complexity of $O(nS + bT + bdT)$, similar to ZO-SVRG in terms of $nS$ but replacing the $ndS$ full gradient cost of FO-SVRG with a cheaper $nS$ ZO estimation cost for snapshots, while achieving a fast $O(1/T + 1/b)$ convergence rate. This makes its computational complexity significantly slower than FO-SVRG, especially when $d$ is large. The VAMO (multi-point) variant has a complexity of $O(qnS + qbT + bdT)$. Here, increasing $q$ (the number of ZO sampling directions) leads to higher complexity but also improves the convergence rate to $O(1/T + (1 - q/d)^2/b)$, reducing the $O(1/b)$ error term and making its performance more comparable to FO-SVRG, particularly if $q \ll d$. This demonstrates that our proposed methods provide a flexible and often more efficient trade-off between computational cost and convergence performance compared to existing pure FO or ZO approaches. Our work further develops such an adaptive hybrid approach by specifically integrating ZO estimation within the SVRG structure, aiming to reduce the full gradient cost while preserving strong convergence guarantees independent of the parameter dimension.

## D    ZO GRADIENT ESTIMATOR

**Lemma 1.** *Under the assumptions in Section 3.1, and define $f_\mu = \mathbb{E}_{\mathbf{u} \sim U_b}[f(\mathbf{x} + \mu \mathbf{u})]$ where $U_b$ is the uniform distribution over the unit Euclidean ball. Then:*

*(i) $f_\mu$ is L-smooth with*

$$\nabla f_\mu(\mathbf{x}) = \mathbb{E}_{\mathbf{u}}\big[\hat{\nabla} f(\mathbf{x})\big]. \tag{12}$$

*(ii) For any $\mathbf{x} \in \mathbb{R}^d$:*

$$|f_\mu(\mathbf{x}) - f(\mathbf{x})| \le \frac{L\mu^2}{2}, \tag{13}$$

$$\|\nabla f_\mu(\mathbf{x}) - \nabla f(\mathbf{x})\|_2^2 \le \frac{\mu^2 L^2 d^2}{4}, \tag{14}$$

$$\frac{1}{2}\|\nabla f(\mathbf{x})\|_2^2 - \frac{\mu^2 L^2 d^2}{4} \le \|\nabla f_\mu(\mathbf{x})\|_2^2 \le 2\|\nabla f(\mathbf{x})\|_2^2 + \frac{\mu^2 L^2 d^2}{2}. \tag{15}$$

*(iii) For any $\mathbf{x} \in \mathbb{R}^d$:*

$$\mathbb{E}_{\mathbf{u}}\big[\|\hat{\nabla} f(\mathbf{x}) - \nabla f_\mu(\mathbf{x})\|_2^2\big] \le 2d\|\nabla f(\mathbf{x})\|_2^2 + \frac{\mu^2 L^2 d^2}{2}. \tag{16}$$

*Proof.* See the proof of Lemma 1 in (Liu et al., 2018b) □

**Lemma 2.** *Under the conditions of Lemma 1:*

*(i) For any $\mathbf{x} \in \mathbb{R}^d$:*

$$\nabla f_\mu(\mathbf{x}) = \mathbb{E}_{\mathbf{u}}\big[\hat{\nabla} f(\mathbf{x})\big]. \tag{17}$$

*where $\hat{\nabla} f(\mathbf{x})$ is the multi-point gradient estimate.*

*(ii) For any $\mathbf{x} \in \mathbb{R}^d$:*

$$\mathbb{E}\big[\|\hat{\nabla} f(\mathbf{x}) - \nabla f_\mu(\mathbf{x})\|_2^2\big] \le \frac{2d}{q}\|\nabla f(\mathbf{x})\|_2^2 + \frac{\mu^2 L^2 d^2}{2q}. \tag{18}$$

*Proof.* See the proof of Lemma 2 in (Liu et al., 2018b) □

## E    SECOND-ORDER MOMENT OF THE HYBRID GRADIENT ESTIMATOR

The primary goal of our convergence analysis is to establish theoretical guarantees for VAMO in solving non-convex optimization problems. Specifically, we aim to bound the expected squared norm of the gradient, $\mathbb{E}[\|\nabla f(\bar{\mathbf{x}})\|_2^2]$, as shown in Theorem 1. Due to the hybrid structure of the gradient estimator $\mathbf{v}_k^s$ used in VAMO, directly analyzing the final convergence metric is challenging. As a key intermediate step, we first derive an upper bound on the second-order moment $\mathbb{E}[\|\mathbf{v}_k^s\|_2^2]$.

**Proposition 1.** *Under the assumptions in Section 3.1, and two-point ZO gradient estimate is used in Algorithm 1. The blended gradient $\mathbf{v}_k^s$ in Step 7 of Algorithm 1 satisfies,*

$$\mathbb{E}\big[\|\mathbf{v}_k^s\|_2^2\big] \le 4\bigg(2\alpha^2 - 2\alpha + 1 + \frac{24d\delta_n}{b}\alpha^2\bigg)\mathbb{E}\big[\|\nabla f(\mathbf{x}_k^s)\|_2^2\big]$$

$$+ \frac{12\delta_n(4d+1)L^2}{b}\alpha^2\mathbb{E}\big[\|\mathbf{x}_0^s - \mathbf{x}_k^s\|_2^2\big] \tag{19}$$

$$+ \frac{9\delta_n}{b}d^2 L^2 \mu^2 \alpha^2 + \frac{4\sigma^2}{b}\bigg(24d\delta_n\alpha^2 + (1-\alpha)^2\bigg),$$

*where $\delta_n = 1$ if the mini-batch contains i.i.d. samples from $[n]$ with replacement, and $\delta_n = I(b < n)$ if samples are randomly selected without replacement. Here $I(b < n)$ is 1 if $b < n$, and 0 if $b = n$.*

*Proof.* In Algorithm 1, we recall that the mini-batch $\mathcal{I}$ is chosen uniformly randomly (with replacement). It is known from Lemma 1 and Lemma 3 that

$$\mathbb{E}_{\mathcal{I}_k} \left[ \nabla f_{\mathcal{I}_k}(\mathbf{x}_k^s) - \hat{\nabla} f_{\mathcal{I}_k}(\mathbf{x}_0^s) \right] = \nabla f(\mathbf{x}_k^s) - \hat{\nabla} f(\mathbf{x}_0^s). \tag{20}$$

We then rewrite $\mathbf{v}_k^s$ as

$$\begin{aligned}
\mathbf{v}_k^s = {} & (1-\alpha)\, \nabla f_{\mathcal{I}_k}(\mathbf{x}_k^s) \\
& + \alpha \left( \nabla f_{\mathcal{I}_k}(\mathbf{x}_k^s) - \hat{\nabla} f_{\mathcal{I}_k}(\mathbf{x}_0^s) - \mathbb{E}_{\mathcal{I}_k} \left[ \nabla f_{\mathcal{I}_k}(\mathbf{x}_k^s) - \hat{\nabla} f_{\mathcal{I}_k}(\mathbf{x}_0^s) \right] + \nabla f(\mathbf{x}_k^s) \right)
\end{aligned} \tag{21}$$

Taking the expectation of $\|\mathbf{v}_k^s\|_2^2$ with respect to all the random variables, we have

$$\begin{aligned}
\mathbb{E}\left[ \|\mathbf{v}_k^s\|_2^2 \right] \leq {} & 2\,(1-\alpha)^2\, \mathbb{E}\left[ \|\nabla f_{\mathcal{I}_k}(\mathbf{x}_k^s)\|_2^2 \right] \\
& + 2\alpha^2 \mathbb{E}\left[ \left\| \nabla f_{\mathcal{I}_k}(\mathbf{x}_k^s) - \hat{\nabla} f_{\mathcal{I}_k}(\mathbf{x}_0^s) - \mathbb{E}_{\mathcal{I}_k} \left[ \nabla f_{\mathcal{I}_k}(\mathbf{x}_k^s) - \hat{\nabla} f_{\mathcal{I}_k}(\mathbf{x}_0^s) \right] + \nabla f(\mathbf{x}_k^s) \right\|_2^2 \right] \\
\leq {} & 4\alpha^2 \mathbb{E}\left[ \left\| \nabla f_{\mathcal{I}_k}(\mathbf{x}_k^s) - \hat{\nabla} f_{\mathcal{I}_k}(\mathbf{x}_0^s) - \mathbb{E}_{\mathcal{I}_k} \left[ \nabla f_{\mathcal{I}_k}(\mathbf{x}_k^s) - \hat{\nabla} f_{\mathcal{I}_k}(\mathbf{x}_0^s) \right] \right\|_2^2 \right] \\
& + 4\alpha^2 \mathbb{E}\left[ \|\nabla f(\mathbf{x}_k^s)\|_2^2 \right] + 2\,(1-\alpha)^2\, \mathbb{E}\left[ \|\nabla f_{\mathcal{I}_k}(\mathbf{x}_k^s)\|_2^2 \right]
\end{aligned} \tag{22}$$

where the first inequality holds due to Lemma 4. Based on equation 20, we note that the following holds

$$\begin{aligned}
& \sum_{i=1}^n \left\{ \nabla f_i(\mathbf{x}_k^s) - \hat{\nabla} f_i(\mathbf{x}_0^s) - \mathbb{E}_{\mathcal{I}_k} \left[ \nabla f_{\mathcal{I}_k}(\mathbf{x}_k^s) - \hat{\nabla} f_{\mathcal{I}_k}(\mathbf{x}_0^s) \right] \right\} \\
& = n(\nabla f(\mathbf{x}_k^s) - \hat{\nabla} f(\mathbf{x}_0^s)) - n(\nabla f(\mathbf{x}_k^s) - \hat{\nabla} f(\mathbf{x}_0^s)) = 0.
\end{aligned} \tag{23}$$

Based on equation 23 and applying Lemma 1 and Lemma 3, the first term at the right hand side (RHS) of equation 22 yields

$$\begin{aligned}
& \mathbb{E}\left[ \left\| \nabla f_{\mathcal{I}_k}(\mathbf{x}_k^s) - \hat{\nabla} f_{\mathcal{I}_k}(\mathbf{x}_0^s) - \mathbb{E}_{\mathcal{I}_k} \left[ \nabla f_{\mathcal{I}_k}(\mathbf{x}_k^s) - \hat{\nabla} f_{\mathcal{I}_k}(\mathbf{x}_0^s) \right] \right\|_2^2 \right] \\
& \leq \frac{\delta_n}{bn} \sum_{i=1}^n \mathbb{E}\left[ \|\nabla f_i(\mathbf{x}_k^s) - \hat{\nabla} f_i(\mathbf{x}_0^s) - (\nabla f(\mathbf{x}_k^s) - \hat{\nabla} f(\mathbf{x}_0^s))\|_2^2 \right] \\
& = \mathbb{E}\left[ \frac{\delta_n}{b} \left( \frac{1}{n} \sum_{i=1}^n \|\nabla f_i(\mathbf{x}_k^s) - \hat{\nabla} f_i(\mathbf{x}_0^s)\|_2^2 - \|\nabla f(\mathbf{x}_k^s) - \hat{\nabla} f(\mathbf{x}_0^s)\|_2^2 \right) \right] \\
& \leq \frac{\delta_n}{bn} \sum_{i=1}^n \mathbb{E}\left[ \left\| \nabla f_i(\mathbf{x}_k^s) - \hat{\nabla} f_i(\mathbf{x}_0^s) \right\|_2^2 \right].
\end{aligned} \tag{24}$$

where the first inequality holds due to Lemma 1 and Lemma 3 (taking the expectation with respect to mini-batch $\mathcal{I}$), we define $\delta_n$ as

$$\delta_n = \begin{cases} 1 & \text{if } \mathcal{I} \text{ contains i.i.d. samples with replacement (Lemma 3)} \\ I(b<n) & \text{if } \mathcal{I} \text{ contains samples without replacement (Lemma 4).} \end{cases} \tag{25}$$

Substituting equation 24 into equation 22, we obtain

$$\begin{aligned}
\mathbb{E}\left[ \|\mathbf{v}_k^s\|_2^2 \right] \leq {} & 2\,(1-\alpha)^2\, \mathbb{E}\left[ \|\nabla f_{\mathcal{I}_k}(\mathbf{x}_k^s)\|_2^2 \right] \\
& + \frac{4\alpha^2 \delta_n}{bn} \sum_{i=1}^n \mathbb{E}\left[ \left\| \nabla f_i(\mathbf{x}_k^s) - \hat{\nabla} f_i(\mathbf{x}_0^s) \right\|_2^2 \right] + 4\alpha^2 \mathbb{E}\left[ \|\nabla f(\mathbf{x}_k^s)\|_2^2 \right].
\end{aligned} \tag{26}$$

Similar to Lemma 1, we introduce a smoothing function $f_{i,\mu}$ of $f_i$, and continue to bound the second term at the right hand side (RHS) of equation 26. This yields

$$\mathbb{E}\left[\|\nabla f_i(\mathbf{x}_k^s) - \hat{\nabla} f_i(\mathbf{x}_0^s)\|_2^2\right]$$

$$\leq 3\mathbb{E}\left[\|\nabla f_i(\mathbf{x}_k^s) - \nabla f_{i,\mu}(\mathbf{x}_k^s)\|_2^2\right] + 3\mathbb{E}\left[\|\nabla f_{i,\mu}(\mathbf{x}_0^s) - \hat{\nabla} f_i(\mathbf{x}_0^s)\|_2^2\right]$$

$$+ 3\mathbb{E}\left[\|\nabla f_{i,\mu}(\mathbf{x}_k^s) - \nabla f_{i,\mu}(\mathbf{x}_0^s)\|_2^2\right]$$

$$\leq 6d\mathbb{E}[\|\nabla f_i(\mathbf{x}_0^s)\|_2^2] + \frac{9}{4}L^2 d^2 \mu^2 + 3\mathbb{E}\left[\|\nabla f_{i,\mu}(\mathbf{x}_k^s) - \nabla f_{i,\mu}(\mathbf{x}_0^s)\|_2^2\right]$$

(27)

Since both $f_i$ and $f_{i,\mu}$ are L-smooth (Lemma 1), we have

$$\mathbb{E}\left[\|\nabla f_{i,\mu}(\mathbf{x}_k^s) - \nabla f_{i,\mu}(\mathbf{x}_0^s)\|_2^2\right] \leq L^2 \mathbb{E}\left[\|\mathbf{x}_k^s - \mathbf{x}_0^s\|_2^2\right],$$

$$\mathbb{E}\left[\|\nabla f_i(\mathbf{x}_0^s)\|_2^2\right] \leq 2\mathbb{E}\left[\|\nabla f_i(\mathbf{x}_0^s) - \nabla f_i(\mathbf{x}_k^s)\|_2^2\right] + 2\mathbb{E}\left[\|\nabla f_i(\mathbf{x}_k^s)\|_2^2\right]$$

$$\leq 2L^2 \mathbb{E}\left[\|\mathbf{x}_0^s - \mathbf{x}_k^s\|_2^2\right] + 2\mathbb{E}\left[\|\nabla f_i(\mathbf{x}_k^s)\|_2^2\right].$$

(28)

We obtain

$$\mathbb{E}\left[\|\nabla f_i(\mathbf{x}_k^s) - \hat{\nabla} f_i(\mathbf{x}_0^s)\|_2^2\right]$$

$$\leq 12d\mathbb{E}[\|\nabla f_i(\mathbf{x}_k^s)\|_2^2] + (12d+3)L^2 \mathbb{E}\left[\|\mathbf{x}_0^s - \mathbf{x}_k^s\|_2^2\right] + \frac{9}{4}L^2 d^2 \mu^2$$

$$\leq 24d\mathbb{E}\left[\|\nabla f_i(\mathbf{x}_k^s) - \nabla f(\mathbf{x}_k^s)\|_2^2\right] + 24d\mathbb{E}\left[\|\nabla f(\mathbf{x}_k^s)\|_2^2\right]$$

$$+ (12d+3)L^2 \mathbb{E}\left[\|\mathbf{x}_0^s - \mathbf{x}_k^s\|_2^2\right] + \frac{9}{4}L^2 d^2 \mu^2$$

$$\leq 24d\sigma^2 + 24d\mathbb{E}\left[\|\nabla f(\mathbf{x}_k^s)\|_2^2\right] + (12d+3)L^2 \mathbb{E}\left[\|\mathbf{x}_0^s - \mathbf{x}_k^s\|_2^2\right] + \frac{9}{4}L^2 d^2 \mu^2,$$

(29)

where the last inequality holds due to Assumption in Section 3.1.

We bound the first term at the right hand side (RHS) of equation 26. This yields

$$\mathbb{E}\left[\|\nabla f_{\mathcal{I}_k}(\mathbf{x}_k^s)\|_2^2\right] \leq 2\mathbb{E}\left[\|\nabla f_{\mathcal{I}_k}(\mathbf{x}_k^s) - \nabla f(\mathbf{x}_k^s)\|_2^2\right] + 2\mathbb{E}\left[\|\nabla f(\mathbf{x}_k^s)\|_2^2\right]$$

$$\leq \frac{2}{b}\sigma^2 + 2\mathbb{E}\left[\|\nabla f(\mathbf{x}_k^s)\|_2^2\right]$$

(30)

Therefore, we have

$$\mathbb{E}\left[\|\mathbf{v}_k^s\|_2^2\right] \leq \frac{4(1-\alpha)^2}{b}\sigma^2 + 4(1-\alpha)^2 \mathbb{E}\left[\|\nabla f(\mathbf{x}_k^s)\|_2^2\right]$$

$$+ \frac{12\delta_n(4d+1)L^2}{b}\alpha^2 \mathbb{E}\left[\|\mathbf{x}_0^s - \mathbf{x}_k^s\|_2^2\right] + \left(4 + \frac{96d\delta_n}{b}\right)\alpha^2 \mathbb{E}\left[\|\nabla f(\mathbf{x}_k^s)\|_2^2\right]$$

$$+ \frac{9\delta_n}{b}d^2 L^2 \mu^2 \alpha^2 + \frac{96d\sigma^2 \delta_n}{b}\alpha^2.$$

$$= 4\left(2\alpha^2 - 2\alpha + 1 + \frac{24d\delta_n}{b}\alpha^2\right)\mathbb{E}\left[\|\nabla f(\mathbf{x}_k^s)\|_2^2\right]$$

$$+ \frac{12\delta_n(4d+1)L^2}{b}\alpha^2 \mathbb{E}\left[\|\mathbf{x}_0^s - \mathbf{x}_k^s\|_2^2\right]$$

$$+ \frac{9\delta_n}{b}d^2 L^2 \mu^2 \alpha^2 + \frac{4\sigma^2}{b}\left(24d\delta_n \alpha^2 + (1-\alpha)^2\right).$$

(31)

$\square$

The bound on $\mathbb{E}[\|\mathbf{v}_k^s\|_2^2]$, detailed in Proposition 1, plays a central role in our analysis. It enables us to control the error accumulation during the optimization process and ultimately leads to the convergence rate stated in Theorem 1. Based on Proposition 1, Theorem 1 provides the convergence rate of VAMO in terms of an upper bound on $\mathbb{E}\left[\|\nabla f(\bar{\mathbf{x}})\|_2^2\right]$ at the solution $\bar{\mathbf{x}}$.

## F  PROOF OF THEOREM 1

*Proof.* Since $f$ is L-smooth (Lemma 1), from Lemma 5 we have

$$
f(\mathbf{x}_k^{s+1}) \leq f(\mathbf{x}_k^s) + \langle \nabla f(\mathbf{x}_k^s), \mathbf{x}_k^{s+1} - \mathbf{x}_k^s \rangle + \frac{L}{2}\|\mathbf{x}_k^{s+1} - \mathbf{x}_k^s\|_2^2
$$

$$
= f(\mathbf{x}_k^s) - \eta_k \langle \nabla f(\mathbf{x}_k^s), \mathbf{v}_k^s \rangle + \frac{L}{2}\eta_k^2\|\mathbf{v}_k^s\|_2^2
\tag{32}
$$

where the last equality holds due to $x_{k+1}^s = x_k^s - \eta_k v_k^s$. Since $x_k^s$ and $x_0^s$ are independent of $\mathcal{I}$ and random directions $u$ used for ZO gradient estimates, from equation 12 we obtain

$$
\mathbb{E}_{\mathbf{u},\mathcal{I}_k}\left[\mathbf{v}_k^s\right] = \mathbb{E}_{\mathbf{u},\mathcal{I}_k}\left[\nabla f_{\mathcal{I}_k}(\mathbf{x}_k^s) - \alpha\left(\hat{\nabla} f_{\mathcal{I}_k}(\mathbf{x}_0^s) - \hat{\nabla} f(\mathbf{x}_0^s)\right)\right]
$$

$$
= \nabla f(\mathbf{x}_k^s) - \alpha\left(\nabla f_\mu(\mathbf{x}_0^s) - \nabla f_\mu(\mathbf{x}_0^s)\right) = \nabla f(\mathbf{x}_k^s).
\tag{33}
$$

Combining equation 32 and equation 33, we have

$$
\mathbb{E}\left[f(\mathbf{x}_{k+1}^s)\right] \leq \mathbb{E}\left[f(\mathbf{x}_k^s)\right] - \eta_k \mathbb{E}\left[\|\nabla f(\mathbf{x}_k^s)\|_2^2\right] + \frac{L}{2}\eta_k^2 \mathbb{E}\left[\|\mathbf{v}_k^s\|_2^2\right],
\tag{34}
$$

where the expectation is taken with respect to all random variables.
At RHS of equation 34, the upper bound on $\mathbb{E}\left[\|\mathbf{v}_k^s\|_2^2\right]$ is given by Proposition 1,

$$
\mathbb{E}\left[\|\mathbf{v}_k^s\|_2^2\right] \leq 4\left(2\alpha^2 - 2\alpha + 1 + \frac{24 d \delta_n}{b}\alpha^2\right)\mathbb{E}\left[\|\nabla f(\mathbf{x}_k^s)\|_2^2\right]
$$

$$
+ \frac{12\delta_n(4d+1)L^2}{b}\alpha^2 \mathbb{E}\left[\|\mathbf{x}_0^s - \mathbf{x}_k^s\|_2^2\right]
\tag{35}
$$

$$
+ \frac{9\delta_n}{b}d^2 L^2 \mu^2 \alpha^2 + \frac{4\sigma^2}{b}\left(24 d \delta_n \alpha^2 + (1-\alpha)^2\right).
$$

In equation 34, we further bound $\mathbb{E}\left[\|\mathbf{x}_{k+1}^s - \mathbf{x}_0^s\|_2^2\right]$ as,

$$
\mathbb{E}\left[\|\mathbf{x}_{k+1}^s - \mathbf{x}_0^s\|_2^2\right] = \mathbb{E}\left[\|\mathbf{x}_{k+1}^s - \mathbf{x}_k^s + \mathbf{x}_k^s - \mathbf{x}_0^s\|_2^2\right]
$$

$$
= \eta_k^2 \mathbb{E}\left[\|\mathbf{v}_k^s\|_2^2\right] + \mathbb{E}\left[\|\mathbf{x}_k^s - \mathbf{x}_0^s\|_2^2\right] - 2\eta_k \mathbb{E}\left[\langle \mathbf{v}_k^s, \mathbf{x}_k^s - \mathbf{x}_0^s \rangle\right]
$$

$$
= \eta_k^2 \mathbb{E}\left[\|\mathbf{v}_k^s\|_2^2\right] + \mathbb{E}\left[\|\mathbf{x}_k^s - \mathbf{x}_0^s\|_2^2\right] - 2\eta_k \mathbb{E}\left[\langle \nabla f(\mathbf{x}_k^s), \mathbf{x}_k^s - \mathbf{x}_0^s \rangle\right]
\tag{36}
$$

$$
\leq \eta_k^2 \mathbb{E}\left[\|\mathbf{v}_k^s\|_2^2\right] + \mathbb{E}\left[\|\mathbf{x}_k^s - \mathbf{x}_0^s\|_2^2\right] + 2\eta_k \mathbb{E}\left[\frac{1}{2\beta_k}\|\nabla f(\mathbf{x}_k^s)\|_2^2 + \frac{\beta_k}{2}\|\mathbf{x}_k^s - \mathbf{x}_0^s\|_2^2\right],
$$

We introduce a Lyapunov function with respect to $f_\mu$,

$$
R_k^s = \mathbb{E}\left[f(\mathbf{x}_k^s) + c_k\|\mathbf{x}_k^s - \mathbf{x}_0^s\|_2^2\right],
\tag{37}
$$

for some $c_k > 0$, Substituting equation 34 and equation 36 into $R_{k+1}^s$, we obtain

$$
R_{k+1}^s = \mathbb{E}\left[f(\mathbf{x}_{k+1}^s) + c_{k+1}\|\mathbf{x}_{k+1}^s - \mathbf{x}_0^s\|_2^2\right]
$$

$$
\leq \mathbb{E}\left[f(\mathbf{x}_k^s) - \eta_k\|\nabla f(\mathbf{x}_k^s)\|_2^2 + \frac{L}{2}\eta_k^2\|\mathbf{v}_k^s\|_2^2\right] + \mathbb{E}\left[c_{k+1}\eta_k^2\|\mathbf{v}_k^s\|_2^2 + c_{k+1}\|\mathbf{x}_k^s - \mathbf{x}_0^s\|_2^s\right]
$$

$$
+ \mathbb{E}\left[\frac{c_{k+1}\eta_k}{\beta_k}\|\nabla f(\mathbf{x}_k^s)\|_2^2 + c_{k+1}\beta_k\eta_k\|\mathbf{x}_k^s - \mathbf{x}_0^s\|_2^2\right]
\tag{38}
$$

$$
= \mathbb{E}\left[f(\mathbf{x}_k^s)\right] - \left(\eta_k - \frac{c_{k+1}\eta_k}{\beta_k}\right)\mathbb{E}\left[\|\nabla f(\mathbf{x}_k^s)\|_2^2\right]
$$

$$
+ (c_{k+1} + c_{k+1}\beta_k\eta_k)\mathbb{E}\left[\|\mathbf{x}_k^s - \mathbf{x}_0^s\|_2^2\right] + \left(\frac{L}{2}\eta_k^2 + c_{k+1}\eta_k^2\right)\mathbb{E}\left[\|\mathbf{v}_k^s\|_2^2\right].
$$

Moreover, substituting equation 35 into equation 38, we have

$$
\begin{aligned}
R_{k+1}^s \leq & \mathbb{E}\left[f(\mathbf{x}_k^s)\right] - \left(\eta_k - \frac{c_{k+1}\eta_k}{\beta_k}\right) \mathbb{E}\left[\|\nabla f(\mathbf{x}_k^s)\|_2^2\right] + (c_{k+1} + c_{k+1}\beta_k\eta_k) \mathbb{E}\left[\|\mathbf{x}_k^s - \mathbf{x}_0^s\|_2^2\right] \\
& + \left(\frac{L}{2}\eta_k^2 + c_{k+1}\eta_k^2\right) \frac{12(4d+1)L^2\delta_n}{b}\alpha^2 \mathbb{E}\left[\|\mathbf{x}_k^s - \mathbf{x}_0^s\|_2^2\right] \\
& + 4\left(\frac{L}{2}\eta_k^2 + c_{k+1}\eta_k^2\right)\left(2\alpha^2 - 2\alpha + 1 + \frac{24d\delta_n}{b}\alpha^2\right) \mathbb{E}\left[\|\nabla f(\mathbf{x}_k^s)\|_2^2\right] \\
& + \left(\frac{L}{2}\eta_k^2 + c_{k+1}\eta_k^2\right)\left(\frac{9\delta_n}{b}d^2L^2\mu^2\alpha^2 + \frac{4\sigma^2}{b}\left(24d\delta_n\alpha^2 + (1-\alpha)^2\right)\right).
\end{aligned}
\tag{39}
$$

The definition of $c_k$ is given by

$$
c_k = c_{k+1} + \beta_k\eta_k c_{k+1} + \left(\frac{L}{2}\eta_k^2 + c_{k+1}\eta_k^2\right)\frac{12(4d+1)L^2\delta_n}{b}\alpha^2
\tag{40}
$$

Based on the definition of $c_k$ and the definition of $R_k^s$ in equation 37, we can simplify the inequality equation 39 as

$$
\begin{aligned}
R_{k+1}^s \leq & R_k^s - \left(\eta_k - \frac{c_{k+1}\eta_k}{\beta_k}\right) \mathbb{E}\left[\|\nabla f(\mathbf{x}_k^s)\|_2^2\right] \\
& + 4\left(\frac{L}{2}\eta_k^2 + c_{k+1}\eta_k^2\right)\left(2\alpha^2 - 2\alpha + 1 + \frac{24d\delta_n}{b}\alpha^2\right) \mathbb{E}\left[\|\nabla f(\mathbf{x}_k^s)\|_2^2\right] \\
& + \left(\frac{L}{2}\eta_k^2 + c_{k+1}\eta_k^2\right)\left(\frac{9\delta_n}{b}d^2L^2\mu^2\alpha^2 + \frac{4\sigma^2}{b}\left(24d\delta_n\alpha^2 + (1-\alpha)^2\right)\right) \\
= & R_k^s - \gamma_k \mathbb{E}\left[\|\nabla f(\mathbf{x}_k^s)\|_2^2\right] + \chi_k,
\end{aligned}
\tag{41}
$$

where $\gamma_k$ and $\chi_k$ are coefficients given by

$$
\begin{aligned}
\gamma_k &= \left(1 - \frac{c_{k+1}}{\beta_k}\right)\eta_k - 4\left(\frac{L}{2} + c_{k+1}\right)\left(2\alpha^2 - 2\alpha + 1 + \frac{24d\delta_n}{b}\alpha^2\right)\eta_k^2, \\
\chi_k &= \left(\frac{L}{2} + c_{k+1}\right)\left(\frac{9\delta_n}{b}d^2L^2\mu^2\alpha^2 + \frac{4\sigma^2}{b}\left(24d\delta_n\alpha^2 + (1-\alpha)^2\right)\right)\eta_k^2
\end{aligned}
\tag{42}
$$

Taking a telescopic sum for equation 42, we obtain

$$
R_m^s \leq R_0^s - \sum_{k=0}^{m-1} \gamma_k \mathbb{E}\left[\|\nabla f(\mathbf{x}_k^s)\|_2^2\right] + \chi_m,
\tag{43}
$$

where $\chi_m = \sum_{k=0}^{m-1} \chi_k$. It is known from equation 37 that,

$$
R_0^s = \mathbb{E}\left[f(\mathbf{x}_0^s)\right], \quad R_m^s = \mathbb{E}\left[f(\mathbf{x}_m^s)\right],
\tag{44}
$$

where the last equality used the fact that $c_m = 0$, since $\bar{\mathbf{x}}_{s-1} = \mathbf{x}_0^s$ and $\bar{\mathbf{x}}_s = \mathbf{x}_m^s$, we obtain

$$
R_0^s - R_m^s = \mathbb{E}\left[f(\bar{\mathbf{x}}_{s-1}) - f(\bar{\mathbf{x}}_s)\right].
\tag{45}
$$

Telescoping the sum for $s = 1, 2, \ldots, S$, we obtain,

$$
\sum_{s=1}^{S}\sum_{k=0}^{m-1} \gamma_k \mathbb{E}[\|\nabla f(\mathbf{x}_k^s)\|_2^2] \leq \mathbb{E}[f(\bar{\mathbf{x}}_0) - f(\bar{\mathbf{x}}_S)] + S\chi_m.
\tag{46}
$$

let $\bar{\gamma} = \min_k \gamma_k$ and we choose $\bar{\mathbf{x}}$ uniformly random from $\{\{\mathbf{x}_k^s\}_{k=0}^{m-1}\}_{s=1}^{S}$, then we obtain

$$
\mathbb{E}[\|\nabla f(\bar{\mathbf{x}})\|_2^2] \leq \frac{\mathbb{E}[f(\bar{\mathbf{x}}_0) - f^*]}{T\bar{\gamma}} + \frac{S\chi_m}{T\bar{\gamma}}.
\tag{47}
$$

$\square$

## G  PROOF OF COROLLARY 1

*Proof.* We start by rewriting $c_k$ in equation 40 as

$$c_k = (1+\theta)c_{k+1} + \frac{6(1+4d)L^3\delta_n\eta^2}{b}\alpha^2 \tag{48}$$

where $\theta = \beta\eta + \frac{12(1+4d)L^2\delta_n\eta^2}{b}\alpha^2$. The recursive formula equation 48 implies that $c_k \leq c_0$ for any $k$, and

$$c_0 = \frac{6(1+4d)L^3\delta_n\eta^2\alpha^2}{b}\frac{(1+\theta)^m - 1}{\theta}. \tag{49}$$

Based on the choice of $\eta = \frac{\rho}{L}$, $\alpha = \frac{1}{d}$, and $\beta = L$, we have

$$\theta = \rho + \frac{12(4d+1)\delta_n\rho^2}{bd^2} \tag{50}$$

where we have used the fact that $\delta_n \leq 1$, Substituting equation 50 into equation 49, we have

$$c_k \leq c_0 = \frac{6(1+4d)L^3\delta_n\alpha^2}{b}\frac{\eta^2}{\theta}[(1+\theta)^m - 1] = \frac{6(1+4d)L\rho\delta_n}{bd^2 + 12(4d+1)\delta_n\rho}[(1+\theta)^m - 1]$$

$$\leq \frac{30L\rho\delta_n}{bd}[(1+\theta)^m - 1] \leq \frac{30L\rho\delta_n}{bd}(e - 1) \leq \frac{60L\rho\delta_n}{bd}, \tag{51}$$

where the third inequality holds since $(1+\theta)^m \leq (1 + \frac{31\rho}{d})^m$, $(1 + 1/a)^a \leq \lim_{a\to\infty}(1 + \frac{1}{a})^a = e$ for $a > 0$, and the last inequality loosely uses the notion '$\leq$' since $e < 3$.
We recall from equation 41 that

$$\bar{\gamma} = \min_{0 \leq k \leq m-1}\left\{\left(1 - \frac{c_{k+1}}{\beta_k}\right)\eta_k - 4\left(\frac{L}{2} + c_{k+1}\right)\left(2\alpha^2 - 2\alpha + 1 + \frac{24d\delta_n}{b}\alpha^2\right)\eta_k^2\right\}. \tag{52}$$

Since $\eta_k = \eta$, $\beta_k = \beta$ and $\eta_k = \eta$, $\beta_k = \beta$, we have

$$\bar{\gamma} \geq \left(1 - \frac{c_0}{\beta}\right)\eta - 4\left(\frac{L}{2} + c_0\right)\left(2\alpha^2 - 2\alpha + 1 + \frac{24d\delta_n}{b}\alpha^2\right)\eta^2. \tag{53}$$

From equation 51 and the definition of $\beta$, we have

$$\frac{c_0}{\beta} \leq \frac{60\rho}{bd}, \tag{54}$$

and

$$\left(\frac{L}{2} + c_0\right)\left(2\alpha^2 - 2\alpha + 1 + \frac{24d\delta_n}{b}\alpha^2\right)\eta$$

$$\leq \left(\frac{L}{2} + \frac{60L\rho}{bd}\right)\left(\frac{2}{d^2} - \frac{2}{d} + 1 + \frac{24\delta_n}{bd}\right)\frac{\rho}{L} \tag{55}$$

$$\leq \rho\left(1 + \frac{24}{bd}\right)$$

Substituting equation 54 and equation 55 into equation 53, we obtain

$$\bar{\gamma} \geq \eta\left(1 - \frac{60\rho}{bd} - 4\rho - \frac{96\rho}{bd}\right) \geq \eta\left(1 - \frac{156\rho}{bd} - 4\rho\right), \tag{56}$$

where we have used the fact that $b < d$. Moreover, if we set $\rho \leq \frac{1}{160}$, then $\bar{\gamma} > 0$. In other words, the current parameter setting is valid for Theorem 1. Upon defining a universal constant $z_0 = 1 - \frac{156\rho}{bd} - 4\rho$, we have

$$\bar{\gamma} \geq \eta z_0 \tag{57}$$

Next, we find the upper bound on $\chi_m$ in equation 41 given the current parameter setting and $c_k \leq c_0$,

$$\chi_m \leq m\left(\frac{L}{2} + c_0\right)\left(\frac{9\delta_n}{b}d^2L^2\mu^2\alpha^2 + \frac{4\sigma^2}{b}\left(24d\delta_n\alpha^2 + (1-\alpha)^2\right)\right)\eta^2 \tag{58}$$

Based on $\bar{\gamma} \geq \eta z_0$ and $c_0 \leq 60 L \rho \delta_n \leq \frac{L}{2}$, we have

$$\frac{\chi_m}{\bar{\gamma}} \leq m\rho \left( \frac{9\delta_n}{bz_0} d^2 L^2 \mu^2 \alpha^2 + \frac{4\sigma^2}{bz_0} \left( 24 d \delta_n \alpha^2 + (1-\alpha)^2 \right) \right) \tag{59}$$

since $T = Sm$, and $\mu = \frac{1}{\sqrt{T}}$, the above inequality yields

$$\frac{S\chi_m}{T\bar{\gamma}} \leq \frac{9\rho L^2 \delta_n}{z_0 bT} + \frac{4\sigma^2}{bz_0} \left( \frac{24\delta_n}{d} + (1-\frac{1}{d})^2 \right) = O\left( \frac{1}{Tb} + \frac{1}{b} \right), \tag{60}$$

where in the big $O$ notation, we only keep the dominant terms and ignore the constant numbers that are independent of $d$, $b$, and $T$.

Substituting equation 57 and equation 60 into equation 7, we have

$$\mathbb{E}[\|\nabla f(\bar{\mathbf{x}})\|_2^2] \leq \frac{[f(\bar{\mathbf{x}}_0) - f^*]}{Tz_0} \frac{L}{\rho} + \frac{S\chi_m}{T\bar{\gamma}} = O\left( \frac{1}{T} + \frac{1}{bT} + \frac{1}{b} \right). \tag{61}$$

$\square$

## H  Proof of Theorem 2

*Proof.* Motivated by Proposition 1, we first bound $\|\mathbf{v}_k^s\|_2^2$, Following, we have

$$\mathbb{E}\left[ \|\mathbf{v}_k^s\|_2^2 \right] \leq 2(1-\alpha)^2 \mathbb{E}\left[ \|\nabla f_{\mathcal{I}_k}(\mathbf{x}_k^s)\|_2^2 \right]$$
$$+ \frac{4\alpha^2 \delta_n}{bn} \sum_{i=1}^n \mathbb{E}\left[ \left\| \nabla f_i(\mathbf{x}_k^s) - \hat{\nabla} f_i(\mathbf{x}_0^s) \right\|_2^2 \right] + 4\alpha^2 \mathbb{E}\left[ \|\nabla f(\mathbf{x}_k^s)\|_2^2 \right]. \tag{62}$$

Following together with equation 18, we can obtain that

$$\mathbb{E}\left[ \|\nabla f_i(\mathbf{x}_k^s) - \hat{\nabla} f_i(\mathbf{x}_0^s)\|_2^2 \right]$$
$$\leq \frac{24d}{q}\sigma^2 + \frac{24d}{q}\mathbb{E}\left[ \|\nabla f(\mathbf{x}_k^s)\|_2^2 \right]$$
$$+ \left( 3 + \frac{12d}{q} \right) L^2 \mathbb{E}\left[ \|\mathbf{x}_0^s - \mathbf{x}_k^s\|_2^2 \right] + \left( \frac{3}{4} + \frac{3}{2q} \right) L^2 d^2 \mu^2, \tag{63}$$

Substituting equation 63 and equation 30 into equation 62, we have:

$$\mathbb{E}\left[ \|\mathbf{v}_k^s\|_2^2 \right] \leq 4\left( \left( 2 + \frac{24d\delta_n}{qb} \right) \alpha^2 - 2\alpha + 1 \right) \mathbb{E}\left[ \|\nabla f(\mathbf{x}_k^s)\|_2^2 \right]$$
$$+ \frac{12L^2\delta_n}{b}\left( 1 + \frac{4d}{q} \right)\alpha^2 \mathbb{E}\left[ \|\mathbf{x}_0^s - \mathbf{x}_k^s\|_2^2 \right] + \frac{3\delta_n}{b}\left( 1 + \frac{2}{q} \right) L^2 d^2 \mu^2 \alpha^2 \tag{64}$$
$$+ \frac{4\sigma^2}{b}\left( \frac{24d\delta_n \alpha^2}{q} + (1-\alpha)^2 \right).$$

Substituting equation 64 into equation 38, we have:

$$R_{k+1}^s \leq \mathbb{E}[f(\mathbf{x}_k^s)] - \left( \eta_k - \frac{c_{k+1}\eta_k}{\beta_k} \right) \mathbb{E}\left[ \|\nabla f(\mathbf{x}_k^s)\|_2^2 \right] + (c_{k+1} + c_{k+1}\beta_k\eta_k) \mathbb{E}\left[ \|\mathbf{x}_k^s - \mathbf{x}_0^s\|_2^2 \right]$$

$$+ \left( \frac{L}{2}\eta_k^2 + c_{k+1}\eta_k^2 \right) \frac{12L^2\delta_n}{b}\left( 1 + \frac{4d}{q} \right)\alpha^2 \mathbb{E}\left[ \|\mathbf{x}_k^s - \mathbf{x}_0^s\|_2^2 \right]$$

$$+ 4\left( \frac{L}{2}\eta_k^2 + c_{k+1}\eta_k^2 \right)\left( \left( 2 + \frac{24d\delta_n}{qb} \right)\alpha^2 - 2\alpha + 1 \right) \mathbb{E}\left[ \|\nabla f(\mathbf{x}_k^s)\|_2^2 \right]$$

$$+ \left( \frac{L}{2}\eta_k^2 + c_{k+1}\eta_k^2 \right) \frac{3\delta_n}{b}\left( 1 + \frac{2}{q} \right) L^2 d^2 \mu^2 \alpha^2$$

$$+ \left( \frac{L}{2}\eta_k^2 + c_{k+1}\eta_k^2 \right) \frac{4\sigma^2}{b}\left( \frac{24d\delta_n \alpha^2}{q} + (1-\alpha)^2 \right)$$

$$\tag{65}$$

Based on the definition of $c_k = (1 + \beta_k \eta_k) c_{k+1} + \left(\frac{L}{2} + c_{k+1}\right)\left(1 + \frac{4d}{q}\right)\frac{12L^2\delta_n\eta_k^2\alpha^2}{b}$ and $R_k^s$ given by equation 37, we can simplify equation 65 to

$$
\begin{aligned}
R_{k+1}^s \leq{}& R_k^s - \left(\eta_k - \frac{c_{k+1}\eta_k}{\beta_k}\right)\mathbb{E}\left[\|\nabla f(\mathbf{x}_k^s)\|_2^2\right] \\
&+ 4\left(\frac{L}{2}\eta_k^2 + c_{k+1}\eta_k^2\right)\left(\left(2 + \frac{24d\delta_n}{qb}\right)\alpha^2 - 2\alpha + 1\right)\mathbb{E}\left[\|\nabla f(\mathbf{x}_k^s)\|_2^2\right] \\
&+ \left(\frac{L}{2}\eta_k^2 + c_{k+1}\eta_k^2\right)\frac{3\delta_n}{b}\left(1 + \frac{2}{q}\right)L^2 d^2 \mu^2 \alpha^2 \\
&+ \left(\frac{L}{2}\eta_k^2 + c_{k+1}\eta_k^2\right)\frac{4\sigma^2}{b}\left(\frac{24d\delta_n\alpha^2}{q} + (1-\alpha)^2\right) \\
\leq{}& R_k^s - \gamma_k\mathbb{E}\left[\|\nabla f(\mathbf{x}_k^s)\|_2^2\right] + \chi_k,
\end{aligned}
\tag{66}
$$

where $\gamma_k$ and $\chi_k$ are defined coefficients in Theorem 2.

Based on equation 66 and the following argument in, we can achieve

$$
\mathbb{E}[\|\nabla f(\bar{\mathbf{x}})\|_2^2] \leq \frac{\mathbb{E}[f(\bar{\mathbf{x}}_0) - f^*]}{T\bar{\gamma}} + \frac{S\chi_m}{T\bar{\gamma}}.
\tag{67}
$$

The rest of the proof is similar to the proof of Corollary 1 with the added complexity of the parameter $q$.

Let $\theta = \beta\eta_k + \left(1 + \frac{4d}{q}\right)\frac{12L^2\delta_n\alpha^2}{b}\eta_k^2$, and $c_k = c_{k+1}(1+\theta) + \left(1 + \frac{4d}{q}\right)\frac{6L^3\delta_n\eta_k^2\alpha^2}{b}$. This leads to:

$$
c_0 = \left(1 + \frac{4d}{q}\right)\frac{6L^3\delta_n\eta^2\alpha^2}{b}\frac{(1+\theta)^m - 1}{\theta}
\tag{68}
$$

Let $\eta = \frac{\rho}{L}$, $\alpha = \frac{q}{d}$, $\beta = L$, and $q \leq d$ we have:

$$
\theta = \rho + (q + 4d)\frac{12\delta_n q\rho^2}{bd^2} \leq \rho + 12\rho\left(\frac{q^2}{d^2} + 4\frac{q}{d}\right) \leq \rho + \frac{60\rho q}{d}
\tag{69}
$$

Substituting equation 69 into equation 68, we have:

$$
\begin{aligned}
c_k \leq c_0 ={}& \left(1 + \frac{4d}{q}\right)\frac{6L^3\delta_n\eta^2\alpha^2}{b}\frac{(1+\theta)^m - 1}{\theta} \\
={}& \frac{6(q+4d)L\delta_n\rho q}{bd^2 + 12(q+4d)\delta_n\rho q}\left[(1+\theta)^m - 1\right] \\
\leq{}& \frac{6(q+4d)L\delta_n\rho q}{bd^2}\left[(1+\theta)^m - 1\right] \\
\leq{}& \frac{30L\delta_n\rho q}{bd}(e-1) = \frac{60L\delta_n\rho q}{bd},
\end{aligned}
\tag{70}
$$

where the second inequality holds since $q \leq d$, and the first inequality holds if $m = \lceil\frac{1}{\rho + \frac{108\rho q}{d}}\rceil$

Because we define $\bar{\gamma} = \min_k \gamma_k$, we have

$$
\bar{\gamma} \geq \eta - \frac{c_0\eta}{\beta} - 4\left(\frac{L}{2}\eta^2 + c_0\eta^2\right)\left(\left(2 + \frac{24d\delta_n}{qb}\right)\alpha^2 - 2\alpha + 1\right)
\tag{71}
$$

From equation 70, we have,

$$
\frac{c_0}{\beta} \leq \frac{60\rho q}{bd}
\tag{72}
$$

Because $\eta = \frac{\rho}{L}$, $\alpha = \frac{q}{d}$, and $q \leq d$ we have

$$
\begin{aligned}
&\left(\frac{L}{2}\eta + c_0\eta\right)\left(\left(2 + \frac{24d\delta_n}{qb}\right)\alpha^2 - 2\alpha + 1\right) \\
&\leq \left(\frac{\rho}{2} + \frac{60\rho^2 q}{bd}\right)\left(\frac{2q^2}{d^2} + \frac{24q}{bd} - \frac{2q}{d} + 1\right) \\
&\leq \rho\left(\frac{24q}{bd} + 1\right) \leq \rho\left(\frac{24}{b} + 1\right)
\end{aligned}
\tag{73}
$$

The second inequality holds if we let $\rho \leq \frac{1}{120}$ Substituting equation 73 and equation 72 into equation 71, we can get

$$\bar{\gamma} \geq \eta \left( 1 - \frac{60\rho q}{bd} - 4\rho \left( \frac{24}{b} + 1 \right) \right) = \eta z_0, \tag{74}$$

where $z_0 > 0$, and $\bar{\gamma}$ is a universal constant that is independent of $T$, $b$ and $d$.
Then, we bound $\chi_m = \sum_k \chi_k$

$$
\begin{aligned}
\chi_m \leq{} & m \left( \frac{L}{2}\eta^2 + c_0\eta^2 \right) \frac{3\delta_n}{b} \left( 1 + \frac{2}{q} \right) L^2 d^2 \mu^2 \alpha^2 \\
& + m \left( \frac{L}{2}\eta^2 + c_0\eta^2 \right) \frac{4\sigma^2}{b} \left( \frac{24d\delta_n\alpha^2}{q} + (1 - \alpha)^2 \right)
\end{aligned}
\tag{75}
$$

Because $c_0 \leq \frac{60L\rho q}{bd} \leq \frac{L}{2}$ if $\rho \leq \frac{1}{120}$, this yields

$$
\begin{aligned}
\frac{\chi_m}{\bar{\gamma}} \leq{} & \frac{\rho}{z_0} \frac{3\delta_n}{b} \left( 1 + \frac{2}{q} \right) L^2 \mu^2 q^2 \\
& + \frac{\rho}{z_0} \frac{4\sigma^2}{b} \left( \frac{24q\delta_n}{d} + (1 - \frac{q}{d})^2 \right)
\end{aligned}
\tag{76}
$$

Since $T = Sm$ and $\mu = \frac{1}{q\sqrt{T}}$, we have

$$
\begin{aligned}
\frac{S\chi_m}{T\bar{\gamma}} \leq{} & \frac{\rho}{z_0} \frac{3\delta_n}{b} \left( 1 + \frac{2}{q} \right) \frac{L^2}{T} \\
& + \frac{\rho}{z_0} \frac{4\sigma^2}{b} \left( \frac{24q\delta_n}{d} + (1 - \frac{q}{d})^2 \right) \\
\leq{} & O \left( \frac{1}{bT} + \frac{1}{b} \left( 1 - \frac{q}{d} \right)^2 \right)
\end{aligned}
\tag{77}
$$

Substituting equation 74 and equation 77 into equation 7, we have

$$\mathbb{E}[\|\nabla f(\bar{\mathbf{x}})\|_2^2] \leq \frac{\mathbb{E}[f(\bar{\mathbf{x}}_0) - f^*]}{T z_0} \frac{L}{\rho} + \frac{S\chi_m}{T\bar{\gamma}} = O \left( \frac{1}{T} + \frac{1}{bT} + \frac{1}{b} \left( 1 - \frac{q}{d} \right)^2 \right) \tag{78}$$

$\square$

## H.1 Auxiliary Lemmas

**Lemma 3.** *Let $\{\mathbf{z}_i\}_{i=1}^n$ be a sequence of $n$ vectors. Let $\mathcal{I}$ be a mini-batch of size $b$, which contains i.i.d. samples selected uniformly randomly (with replacement) from $[n]$.*

$$\mathbb{E}_{\mathcal{I}} \left[ \frac{1}{b} \sum_{i \in \mathcal{I}} \mathbf{z}_i \right] = \frac{1}{n} \sum_{j=1}^n \mathbf{z}_j. \tag{79}$$

*When $\sum_{i=1}^n \mathbf{z}_i = \mathbf{0}$, then*

$$\mathbb{E}_{\mathcal{I}} \left[ \left\| \frac{1}{b} \sum_{i \in \mathcal{I}} \mathbf{z}_i \right\|_2^2 \right] = \frac{1}{bn} \sum_{i=1}^n \|\mathbf{z}_i\|_2^2. \tag{80}$$

*Proof.* See the proof of Lemma 4 in (Liu et al., 2018b). $\square$

**Lemma 4.** *Let $\{\mathbf{z}_i\}_{i=1}^n$ be a sequence of $n$ vectors. Let $\mathcal{I}$ be a uniform random mini-batch of $[n]$ with size $b$ (no replacement in samples). Then*

$$\mathbb{E}_{\mathcal{I}} \left[ \frac{1}{b} \sum_{i \in \mathcal{I}} \mathbf{z}_i \right] = \frac{1}{n} \sum_{j=1}^n \mathbf{z}_j. \tag{81}$$

*When $\sum_{i=1}^{n} \mathbf{z}_i = \mathbf{0}$, then*

$$\mathbb{E}_{\mathcal{I}}\left[\left\|\frac{1}{b}\sum_{i\in\mathcal{I}}\mathbf{z}_i\right\|_2^2\right] = \frac{\mathcal{I}(b<n)}{bn}\sum_{i=1}^{n}\|\mathbf{z}_i\|_2^2. \tag{82}$$

*where I is an indicator function, which is equal to 1 if $b < n$ and 0 if $b = n$.*

*Proof.* See the proof of Lemma A.1 in (Lei et al., 2017). □

**Lemma 5.** *For variables $\{\mathbf{z}_i\}_{i=1}^{n}$, we have*

$$\left\|\sum_{i=1}^{n}\mathbf{z}_i\right\|_2^2 \leq n\sum_{i=1}^{n}\|\mathbf{z}_i\|_2^2. \tag{83}$$

*Proof.* See the proof of Lemma 6 in (Liu et al., 2018b). □

**Lemma 6.** *if f is L-smooth, then for any $\mathbf{x}, \mathbf{y} \in \mathbb{R}^d$*

$$|f(\mathbf{x}) - f(\mathbf{y}) - \langle\nabla f_i(\mathbf{y}), \mathbf{x}-\mathbf{y}\rangle| \leq \frac{L}{2}\|\mathbf{x}-\mathbf{y}\|_2^2. \tag{84}$$

*Proof.* This is a direct consequence of Lemma A.2 in (Lei et al., 2017). □

# I  ANALYSIS OF ZEROTH-ORDER GRADIENT ESTIMATION ERROR

This section details bounds on the expected squared error of the ZO gradient estimators used in our work. We consider a ZO gradient estimator $\hat{\nabla}f_i(\mathbf{x})$ for a component function $f_i(\mathbf{x})$, which approximates the true gradient $\nabla f_i(\mathbf{x})$ with an estimation error $\omega_i(\mathbf{x})$, such that $\hat{\nabla}f_i(\mathbf{x}) = \nabla f_i(\mathbf{x}) + \omega_i(\mathbf{x})$. The characteristics of the expected squared error, $\mathbb{E}[\|\omega_i(\mathbf{x})\|_2^2]$, are presented below.

For the two-point ZO gradient estimator of $f_i(\mathbf{x})$, as defined in Equation equation 2 in the main text, the expected squared error is bounded by:

$$\mathbb{E}[\|\omega_i(\mathbf{x})\|_2^2] \leq O(d)\|\nabla f_i(\mathbf{x})\|_2^2 + O(\mu^2 L^2 d^2). \tag{85}$$

Here, $d$ is the problem dimension, $\mu$ is the smoothing parameter, and $L$ is the smoothness constant associated with $f_i$.

Subsequently, for the multi-point ZO gradient estimator of $f_i(\mathbf{x})$ using $2q$ query points, as defined in Equation equation 3 in the main text, the expected squared error is bounded by:

$$\mathbb{E}[\|\omega_i(\mathbf{x})\|_2^2] \leq O(d/q)\|\nabla f_i(\mathbf{x})\|_2^2 + O(\mu^2 L^2 d^2). \tag{86}$$

The detailed proofs for these bounds can be found in Proposition 2 of (Liu et al., 2018b).

# J  EXPERIMENT SETUP

## J.1  ADAPTABILITY EXPERIMENT

The primary objective of this adaptability experiment was to empirically investigate the impact of the number of ZO query points ($q$) on the performance of VAMO and to validate the theoretical benefits of its multi-point ZO estimation strategy (see Section 5). The optimization problem was a finite-sum non-convex least-squares objective: $f(\mathbf{x}) = \frac{1}{n}\sum_{i=1}^{n}(h(\mathbf{x}; \mathbf{z}_i) - y_i)^2$. We configured this synthetic task with $n = 1000$ individual component functions and a parameter dimension of $d = 100$. The function $h(\mathbf{x}; \cdot)$ was parameterized using a simple neural network with a non-convex activation function to ensure the overall non-convexity of the loss landscape.

In this setup, VAMO variants utilizing $q \in \{1, 3, 5\}$ query directions for the multi-point ZO gradient estimator were compared against the classical FO-SGD algorithm. A mini-batch size of $b = 8$ was consistently applied across all methods. Learning rates for both VAMO (for each $q$ setting) and

FO-SGD were individually tuned by selecting the best performing value from the range $[10^{-2}, 10^{-1}]$. For all VAMO variants, the ZO smoothing parameter was fixed at $\mu = 10^{-3}$. The mixing coefficient $\alpha$ for VAMO was also tuned for each value of $q$, guided by the theoretical insights on balancing FO and ZO information discussed in Appendix K.3.

### J.2 MNIST CLASSIFICATION TASK

For the MNIST multi-class image classification task (LeCun et al., 1998), we trained a Multi-Layer Perceptron (MLP) to evaluate VAMO against established baselines. The MLP architecture consisted of an input layer receiving flattened $28 \times 28$ pixel images (784 dimensions), followed by two hidden layers with 32 and 16 units respectively, both employing ReLU activation functions. The final output layer comprised 10 units corresponding to the digit classes, and the network was trained using a standard cross-entropy loss function. Images were normalized to the range $[0, 1]$.

Our proposed VAMO algorithm, configured with a single ZO query direction ($q = 1$), was benchmarked against pure first-order FO-SGD (Robbins & Monro, 1951) and pure zeroth-order methods, ZO-SGD and ZO-SVRG (Liu et al., 2018b; Ghadimi & Lan, 2013). For all methods, the mini-batch size was set to $b = 4$. The learning rates were independently tuned for each method, selected from the range $[10^{-4}, 10^{-3}]$ for the ZO methods, and $[10^{-3}, 10^{-2}]$ for FO methods and VAMO. For VAMO with $q = 1$, we fixed the mixing coefficient at $\alpha = 0.1$ and used a ZO smoothing parameter of $\mu = 10^{-3}$. All models were trained for 10 epochs.

### J.3 FINE-TUNING EXPERIMENTS

To further examine VAMO in realistic large-scale scenarios, we conducted fine-tuning experiments on three language models under the MultiNLI (MNLI) dataset (Williams et al., 2017) for a three-way natural language inference task. In all settings, the training and validation sets were subsampled to 256 and 128 examples, respectively, with a maximum input sequence length of 128 tokens. All models were fine-tuned for 1500 epochs with full precision (FP32) on a single NVIDIA RTX5880 49GB GPU. All experiments involved full-parameter fine-tuning.

For fair comparisons, VAMO was compared against representative FO methods including FO-SGD (Robbins & Monro, 1951), FO-Adagrad (Duchi et al., 2011) and FO-Adam (Kingma & Ba, 2014), and ZO methods including ZO-SGD (Ghadimi & Lan, 2013) and ZO-SVRG (Liu et al., 2018b). We set the batch size as 32. The learning rates of FO methods were tuned within $[1e - 4, 1e - 3]$, while those of ZO methods were tuned within $[1e - 6, 1e - 5]$. For VAMO, the learning rate was also tuned in the range $[1e - 4, 1e - 3]$. We adopted the same learning rate schedule as illustrated in Algorithm 2. For ZO methods, we adopted a smoothing parameter $\mu = 1e - 3$ and $q = 1$ (two-point version). VAMO was configured with the the smoothing parameter $\mu = 1e - 3$, $q = 1$, and inner loop length $m = 10$.

The main difference is in setting the mixing coefficient $\alpha$. $\alpha$ was set to $1e - 2$ for the GPT-2 experiments, while a lower value $\alpha = 5e - 3$ was used for RoBERTa-Large and GPT-2 Medium. We used a smaller $\alpha$ for larger models because the ZO gradient estimates exhibit higher inherent error in higher-dimensional models, so reducing $\alpha$ limits the contribution of the noisier ZO correction and helps stabilize the training. A detailed analysis of $\alpha$ selection and the error of the ZO estimation is provided in Appendix K.3 and Appendix I.

---

**Algorithm 2** Learning Rate Scheduling for VAMO

1: **Input:** Learning rates $\eta_1, \eta_2$, annealing factor $\delta_\eta$, losses $L$, annealing threshold $\kappa$, total number of batches in an epoch $w$
2: Compute moving averages:
3: $L_1 \leftarrow \text{mean}(L[-w, :])$
4: $L_2 \leftarrow \text{mean}(L[-2w, -w])$
5: **if** $\frac{L_1}{L_2} > \kappa$ **then**
6: $\quad \eta_1 \leftarrow \frac{\eta_1}{\delta_\eta}, \quad \eta_2 \leftarrow \frac{\eta_2}{\delta_\eta}$
7: **end if**
8: **Return:** updated $\eta_1, \eta_2$

---

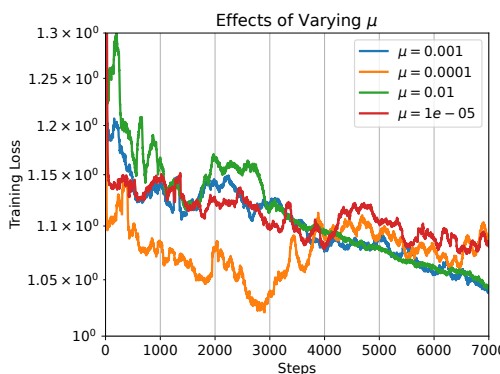

Figure 3: The effects of the smoothing parameter $\mu$ on the performance of VAMO for fine-tuning GPT-2 on MNLI.

## K    ABLATION ON KEY HYPERPARAMETERS

### K.1    ROLE OF $\mu$ (SMOOTHING PARAMETER)

We investigated the role of the parameter $\mu$ in VAMO. Recall that $\mu$ defines as smoothing parameter when computing ZO gradients equation 2 and equation 3. It is known from Spall (1992) that the ZO estimator is asymptotically unbiased as $\mu \to 0$. We wanted to see the practical effects of different $\mu$ settings for VAMO. Therefore we conducted the ablation experiment that the smoothing parameter $\mu$ varied. We fine-tune the GPT-2 (Radford et al., 2019) on the MNLI dataset (Williams et al., 2017). We fixed the key parameters ($\alpha = 0.01$, $q = 1$, and $m = 10$), and the learning rate $\eta$ is set as $1e-3$.

Fig. 3 shows how different values of $\mu$ affect the training loss of VAMO algorithm. We observe that there is no noticeable difference if the value of $\mu$ is sufficient small. Similar findings were also empirically stated in (Malladi et al., 2023) and Gautam et al. (2024). Therefore, in the fine-tuning experiments (see Section 6), we chose the default value of $\mu = 1e-3$.

### K.2    ROLE OF $m$

The parameter $m$ is a significant role in VAMO as it governs the frequency of the full-batch ZO gradient updates. Specifically, a smaller $m$ means that the full-batch ZO gradient estimate is computed more frequently, which can lead to a more effective reduction in gradient variance. To better understand the trade-off between the effectiveness of these updates and their computational overhead, we perform an ablation study on $m$. We consider the task of fine-tuning the GPT-2 (Radford et al., 2019) model on the MNLI dataset (Williams et al., 2017) . For this study, we fixed the other key hyperparameters ($\alpha = 0.1$, $q = 1$, and $\mu = 1e-3$) while varying $m$ and the learning rate $\eta$ is set as $1e-3$.

Fig. 4 shows that there is no noticeable difference when computing the full-batch ZO gradient frequently enough (e.g. $2 \leq m \leq 10$). However, the larger $m$ (e.g. $m \geq 20$) results in diverging behavior. If the value of $m$ is too small, computing the full-batch ZO gradient too frequently will bring additional and unnecessary computational cost. Thus, we chose the default value of $m = 10$ in the fine-tuning experiment.

### K.3    ROLE OF $\alpha$ (MIXING COEFFICIENT)

**Theoretical Analysis.**    The mixing coefficient $\alpha$ in the VAMO update in  equation 6 critically balances the FO stochastic gradient $\nabla f_{\mathcal{I}}(\mathbf{x})$ against the ZO variance correction term $\hat{\nabla} f_{\mathcal{I}}(\hat{\mathbf{x}}) - \hat{\nabla} f(\hat{\mathbf{x}})$. The optimal choice for $\alpha$ directly depends on the estimation error $\omega_i$ inherent in the ZO gradient components ($\hat{\nabla} f_i(\mathbf{x}) = \nabla f_i(\mathbf{x}) + \omega_i$). As established in the literature (Liu et al., 2018b; 2020) and detailed in Appendix I, the expected squared ZO error $\mathbb{E}[\|\omega_i\|_2^2]$ typically scales as $\mathcal{O}(d)$ for two-point estimates and $\mathcal{O}(d/q)$ for multi-point estimates using $q$ random directions. This relationship dictates

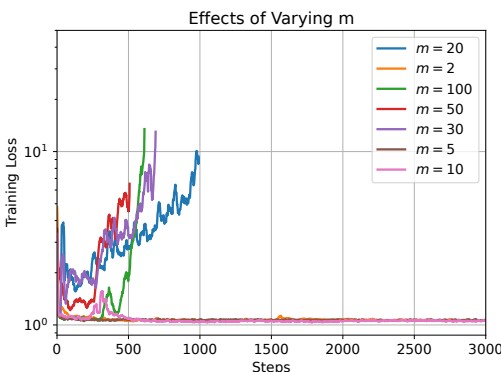

Figure 4: The effects of the frequency of full-batch gradient computation $m$ on the performance of VAMO for fine-tuning GPT-2 on MNLI.

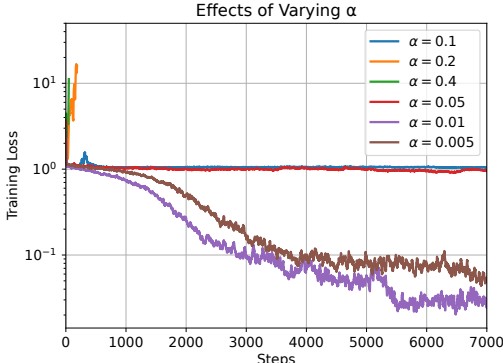

Figure 5: The effects of the mixing coefficient $\alpha$ on the performance of VAMO for fine-tuning GPT-2 on MNLI.

that $\alpha$ should reflect the trustworthiness of the ZO estimates: when the ZO error is substantial (e.g., large $d$, small $q$), a smaller $\alpha$ is warranted to prevent amplifying this error. Conversely, when ZO estimates are more reliable (e.g., larger $q$ reducing error), a larger $\alpha$ can more aggressively leverage the variance reduction. This principled inverse relationship between ZO error magnitude (influenced by $d$ and $q$) and the appropriate scale of $\alpha$ is key. While specific forms like $\alpha \propto 1/d$ or $\alpha \propto q/d$ analyzed in our theoretical sections (e.g., Corollary 1 and Theorem 2) illustrate this adaptive trend, the core insight is that $\alpha$ must be adjusted to counterbalance the ZO estimator's error profile. Such adaptability enables VAMO to effectively navigate the trade-off between computational cost and convergence performance, a central aspect of its practical utility.

**Empirical Results Analysis.** The mixing coefficient $\alpha$ controls how much weight VAMO assigns to the ZO-based variance reduction term relative to the FO gradient. A larger $\alpha$ strengthens the effect of variance correction but also amplifies the noise of the ZO estimator. To study this trade-off, we fine-tuned GPT-2 (Radford et al., 2019) on the MNLI dataset (Williams et al., 2017), fixing other hyperparameters ($m = 10$, $q = 1$, and $\mu = 1e - 3$) while varying $\alpha$ and the learning rate $\eta$ is set as $1e - 3$. As shown in Fig. 5, smaller values (e.g. $\alpha = 1e - 2$) provide a good balance, leading to fast convergence without being overly sensitive to noisy ZO gradients. However, if $\alpha$ is set too small (e.g., $\alpha = 5e - 3$), the variance reduction term becomes underutilized, diminishing its benefit. In contrast, setting $\alpha$ too high (e.g., $\alpha = 0.2$) magnifies estimation error, results in diverging behavior. These results align with our theoretical analysis, which highlights the need for careful tuning of $\alpha$ in high-dimensional settings.

