# OpenReview forum: "VAMO: Efficient Zeroth-Order Variance Reduction for SGD with Faster Convergence"
_ICLR.cc/2026/Conference — ICLR 2026 Conference Withdrawn Submission_

### Official Review · Reviewer_zY9f · 2025-10-30

**Soundness:** 2
**Presentation:** 2
**Contribution:** 2
**Rating:** 4
**Confidence:** 3

**Summary:**

The paper proposes VAMO - a variance reduction optimization method that combines both zero-order gradient estimates and gradient to produce the update direction. Concretely, the algorithm uses the full batch and batch gradient estimates at the checkpoint (obtained from the standard zero-order method) and combines with the batch gradient at the current iterate ($g = \nabla f(x, B) - \alpha (\hat{\nabla} f (x_{\text{cpt}}, B) - \hat{\nabla}f(x_{\text{cpt}}))$). This gives a trade off between convergence and computational complexity compared with ZO-SVRG. In terms of convergence, the rate of VAMO is $1/T + 1/b$ where $b$ is the batch size. This has the extra  $1/b$ term is inherited from ZO-SVRG due to the bias in the estimator, but the first term is improved by a factor $d$. The computational complexity on the other hand is increased from $nS+bT$ to $nS+dbT$ (due to the gradient computation). An extension to this method is to use multi-point query, which can trade off between the improvement in the additional error term and the computational complexity. Experiments show that VAMO has faster convergence than other ZO methods and better memory footprint than FO methods.

**Strengths:**

- Combining FO and ZO in SVRG is an interesting idea and the FO order component improves the performance of the algorithm compared with other ZO methods.
- The presentation of the paper is easy to follow with detailed comparison with prior works.
- Proofs seem all good.

**Weaknesses:**

I have several concerns below.
- I'm not sure I understand the memory analysis in Section 4.3, B.1 and Table 3.
  - For VAMO, why is the memory for the optimizer states only $|x|$? The algorithm needs to store $\hat{\nabla}(x_{\text{cpt}})$ but also $x_{\text{cpt}}$ to compute the estimate $\hat{\nabla}(x_{\text{cpt}}, B)$ (see also line 3-4 in Alg. 1). I don't understand how to reduce the memory to only $|x|$? If my understanding is true, I don't see an improvement in the memory for the proposed method.
  - For Adagrad, why is the optimizer states $2|x|$? Doesn't the algorithm only store the accumulation of the gradients per coordinate so the memory needed is only $|x|$? Same for Adam, shouldn't it be $2|x|$?
- The experiment appears quite weak. First of all, the paper only reports the training loss. While optimization algorithms only optimize the training loss, we also care about the test loss and accuracy. Second, one main motivation the paper mentioned is to overcome limitations of FO and ZO methods for efficient training of large models. However, the set of experiments is quite limited. MNIST is an outdated dataset as an optimization benchmark. Usually, for optimization papers, starting with CIFAR-10/100 will show clearer impacts. I also suggest that the authors use a similar experiment setup in the paper MeZO and add more experiments.

**Questions:**

- See weaknesses.

---

> ### Author Response · Authors · 2025-11-19
>
> We thank the reviewer for their evaluation of this manuscript, but it seems like the reviewer might have overlooked some important aspects of it, and we hope to provide some additional clarifications below.
>
> ---
>
> ### Clarification on Memory Cost of VAMO and Other Optimizers
>
> We first thank the reviewer for pointing out that the optimizer state memory of VAMO in Table 3 should be $2|x|$ , not $|x|$.  VAMO must store both the snapshot weights ($x_0^s$) and the snapshot gradient ($\hat{g}_s$). Additionally, FO-Adam is typically $2|x|$. We will update the table in a revised manuscript.
>
> However we must clarify that **VAMO indeed incurs a lighter memory footprint than most major optimizers while enjoying a speedy convergence**. We point the reviewer to Table 2 of the original manuscript for these results (reproduced in the table below for your perusal), which may have been glossed over. The results show that VAMO's peak memory cost is 32% (RoBERTa-Large) and 26% (GPT-2 Medium) lower than FO-Adam on average. Compared to FO-Adagrad, VAMO's cost is 23% (RoBERTa-Large) and 20% (GPT-2 Medium) lower on average.
>
> Though VAMO has the same optimizer state as FO-Adam, in practical application, in practice, VAMO achieves a substantial improvement in peak memory compared to FO-Adam and Adagrad.  Adaptive optimizers like FO-Adam and Adagrad, for faster computation, rely on non-linear sqrt/div operations, which allocate large, temporary buffers after activations have already filled and fragmented GPU memory, spiking the true peak memory. This "last-minute" allocation often fails to reuse memory efficiently, spiking the true peak memory. VAMO’s update is linear, in-place, and relies on static states. It never requests these problematic temporary buffers, thus avoiding the allocation spike entirely and resulting in a lower practical footprint. This is precisely what theoretical analysis fails to capture. That is why we moved the theoretical table to the appendix, and prioritized the peak memory cost  in Section 6.
>
> **Table 1: Peak GPU Memory Consumption (in GB)**
>
> | Method | RoBERTa-Large (bs = 16) | RoBERTa-Large (bs = 32) | RoBERTa-Large (bs = 64) | GPT-2-Medium (bs = 16) | GPT-2-Medium (bs = 32) | GPT-2-Medium (bs = 64) |
> | --- | --- | --- | --- | --- | --- | --- |
> | FO-SGD | 4.63 | 6.72 | 10.83 | 7.08 | 11.50 | 20.33 |
> | FO-Adagrad | 7.21 | 10.46 | 16.54 | 10.25 | 15.74 | 27.57 |
> | FO-Adam | 8.62 | 11.59 | 17.66 | 11.59 | 17.05 | 29.07 |
> | **VAMO($q=1$)** | **5.95** | **7.83** | **11.96** | **8.40** | **12.62** | **21.46** |

---

> > ### Author Response · Authors · 2025-11-19
> >
> > ### Clarification on Experiment Setup
> >
> > **We believe there is a misunderstanding regarding our experimental scope.** The reviewer stated the experiments were limited to MNIST, and we remind the reviewer to checkout the fine-tuning experiments using different models across different dataset, which is the third part of **Section 6** in the paper. Hopefully it addresses your concern of insufficient experiments.
> >
> > In that section, we explicitly use modern architectures (GPT-2, RoBERTa-Large) and contemporary datasets (MNLI, SST-2) to compare VAMO with baselines. These experiments demonstrate the efficient training of large models using VAMO, which directly validates our core motivation. Furthermore, this experimental setup is consistent with recent work, such as MeZO-SVRG (ICML 2024) [1].
> >
> > Regarding the metric, while training loss is standard for optimization analysis, to address your concern, we now explicitly present the final test accuracies for the "Fine-Tuning Experiments" (from Section 6) in the table below (Table 2). These results correspond to the fine-tuning of RoBERTa-Large, GPT-2 Medium, and GPT-2 Small on the MNLI and SST-2 datasets.
> >
> > As shown in the table, although VAMO converges slower than FO-Adam, it demonstrates better generalization ability in some tasks. For instance, VAMO's test accuracy is 8% higher than that of FO-Adam, in the RoBERTa-Large and 26% higher in the GPT-2 Medium (MNLI) task, while also achieving lower test losses in both. Besides, VAMO has 6% and 13% higher accuracy than that of FO-SGD and FO-Adagrad on average across all tasks, respectively. Compared with ZO methods, VAMO's test accuracy is 11% higher than that of ZO-SGD and ZO-SVRG on average. We believe the improved generalization ability might come from the use of full-batch gradients, since all other algorithms only use mini batch gradients, which may result in insufficient training. We plan to further investigate this phenonmenon in future works.
> >
> > **Table 2: Test Accuracy and Loss**
> > | Model (Dataset) | FO-SGD | FO-Adagrad | FO-Adam | MeZO | ZO-SVRG | **VAMO (Ours)** |
> > | --- | --- | --- | --- | --- | --- | --- |
> > | **RoBERTa-Large** (MNLI) | 75 (0.67) | 72 (0.74) | 61 (0.98) | 73 (0.99) | 70 (1.01) | 78 (0.56) |
> > | **GPT-2 Medium** (MNLI) | 60 (0.90) | 53 (1.33) | 48 (1.03) | 66 (0.89) | 69 (0.88) | 67 (0.85) |
> > | **GPT-2 Small** (SST-2) | 90 (0.42) | 88 (0.37) | 98 (0.03) | 84 (0.59) | 72 (0.64) | 94 (0.16) |
> >
> > ---
> >
> > [1] Gautam, Tanmay, et al. "Variance-reduced zeroth-order methods for fine-tuning language models." *arXiv preprint arXiv:2404.08080* (2024).

---

> > > ### Comment · Reviewer_zY9f · 2025-11-21
> > >
> > > Thank you for the response. I just wanted to clarify the points that I raised in my review (I did not gloss over the experimental results).
> > >
> > > 1. I was aware of the experimental results in Table 2 that the authors reproduced here. My concern was that the theoretical guarantee of the memory requirement in Table 3 is incorrect and does not explain the results in Table 2. In particular, in theory, FO-Adagrad should have better memory requirement and FO-Adam the same as VAMO. Here, I'm not sure I understand the justification for Table 2 you gave in your response. Doesn't it just mean the baselines FO-Adam and FO-Adagrad do not have an implementation that is optimized for the memory, which means the comparison is somewhat unfair?
> > >
> > > 2. I wanted to re-emphasize my concern on the limited set of experiments (and not that the experiments are limited to MNIST). Again, MNIST is outdated and no longer a good benchmark for testing optimization algorithms. For the fine-tuning experiments, there are standard baselines, such as Glue, commonsense datasets. Here the papers only showed the results on 2 datasets, which is insufficient for evaluation. My suggestion was to use the same experimental setup in MeZO paper (Malladi et al.), where results on standard baselines are provided (Table 1 & 2 in this paper). This was not addressed in your response.

---

> > > > ### Author Response · Authors · 2025-11-26
> > > >
> > > > Thank you for your evaluation of this manuscript. And we provide some clarifications and additional experiment results below.
> > > >
> > > > ### Clarification on Memory Analysis
> > > > There is a distinction between theoretical and practical memory cost. The theoretical analysis serves primarily as a reference. For instance, according to the memory analysis by Zhang et al. [1], the optimizer state of FO-Adam is $3|x|$, which is larger than VAMO. In experiments, we prioritized the actual peak memory usage of different methods, and the results show that FO-Adam and FO-Adagrad consume more peak memory usage compared with VAMO.
> > > >
> > > > ---
> > > >
> > > > ### Fine-Tuning Experiments
> > > > To further demonstrate the performance, we further fine-tuned the RoBERTA-Large model on other different datasets, QNLI and COLA.  We now explicitly present the final test accuracies in the Table below. The results further show VAMO's generalization ability. VAMO's test accuracy is 5.5%, 17% and 23% higher than FO-SGD, FO-Adam and FO-Adagrad on average,  respectively. We believe that these additional experiments on RoBERTa-Large are sufficient to demonstrate VAMO's effectiveness and generalization ability compared to standard baselines.
> > > >
> > > > **Table 1: Test Accuracy and Loss**
> > > > | Model (Dataset) | FO-SGD | FO-Adagrad | FO-Adam | **VAMO (Ours)** |
> > > > | --- | --- | --- | --- |--- |
> > > > | **RoBERTa-Large** (QNLI) | 83 (0.47) | 72 (0.65) | 66 (0.71) | 86 (0.39) |
> > > > | **RoBERTa-Large** (COLA) | 81 (0.48) | 75 (0.55) | 75 (0.54) | 87 (0.38) |
> > > >
> > > >
> > > >
> > > > ---
> > > > [1] Zhang, Yihua, et al. "Revisiting zeroth-order optimization for memory-efficient llm fine-tuning: A benchmark." arXiv preprint arXiv:2402.11592 (2024).

---

### Official Review · Reviewer_mDeF · 2025-10-31

**Soundness:** 2
**Presentation:** 3
**Contribution:** 2
**Rating:** 2
**Confidence:** 4

**Summary:**

This paper proposes VAMO, a new stochastic optimization algorithm for large-scale non-convex optimization.
The algorithm aims to bridge the gap between first-order (FO) methods, which have fast
convergence but high computational and memory costs, and zeroth-order (ZO) methods, which are
memory-light but suffer from dimension-dependent and slow convergence.
VAMO is a two-loop "SVRG-style" algorithm. Its central idea is to replace the computationally
expensive full-batch FO gradient (∇f(ˆx)) used in FO-SVRG with a full-batch ZO gradient estimate
(ˆ∇f(ˆx)). The inner-loop update is a novel hybrid estimator: vsk
= ∇fIk (xsk) − α(ˆ∇fIk (ˆx) − ˆ∇f(ˆx)).
This construction cleverly uses a ZO-based variance correction term that has zero expectation, making
vsk an unbiased estimator of the true gradient ∇f(xsk).
The authors provide a theoretical analysis showing the two-point (q = 1) version of VAMO achieves a
convergence rate of O(1/T +1/b), which is dimension-independent, a significant improvement over typical
ZO methods. They also propose a multi-point (q > 1) variant with a rate of O(1/T +(1−q/d)2/b),
which can converge to a stationary point if q = d. The primary claims are that VAMO achieves FOSVRG-
like convergence speed with significantly lower computational and, critically, dynamic memory
costs. Experiments on neural networks and LLM finetuning are presented to support these claims.

**Strengths:**

Novel Hybrid Estimator: The proposed gradient estimator (Eq. 6) is novel. The insight to
use a ZO-based correction term α(ˆ∇fIk (ˆx) − ˆ∇f(ˆx)), which has zero expectation, is technically
sound. This ensures the full estimator vsk is an unbiased estimator of the true current gradient
∇f(xsk), which is an elegant property for the analysis.

2. Dimension-Independent ZO-Hybrid Rate: The theoretical analysis successfully breaks the
dimension-dependency curse of pure ZO methods. Achieving a rate of O(1/T + 1/b) (Corollary
1) that is independent of d is a strong theoretical contribution for a method that incorporates
ZO components.

3. Strong Empirical Performance vs. FO-SGD: The experiments, particularly in Figure 2
(LLM finetuning), show that VAMO (with q = 1) converges significantly faster in terms of both
steps and wall-clock time than FO-SGD. This empirically validates the theoretical advantage of
the O(1/T + ...) rate over FO-SGD’s O(1/√T) rate for reaching a certain precision.

**Weaknesses:**

Despite its theoretical novelty, the paper’s core claims about its practical advantages are based on a
series of critical, and in some cases contradictory, flaws in the analysis of its computational and memory
costs.

1. Fundamentally Misleading Convergence Claims: The paper repeatedly claims an O(1/T )
rate, equating it with FO-SVRG (e.g., Abstract: "significantly improving over SGD’s... rate";
Conclusion: "achieving convergence performance similar to FO-SVRG"). This is a misrepresentation.
The actual derived rate is O(1/T + 1/b). This is not convergence to a stationary point.
It is linear convergence to a noise ball of size O(1/b). This non-vanishing error term, which
dominates at large T, means VAMO (q = 1) cannot achieve high-precision solutions and may, in
fact, converge to a worse solution than FO-SGD (which does converge to ϵ = 0, albeit slower).
This is a crucial distinction that is glossed over.

2. Contradictory and Factually Incorrect Memory Analysis: This is the paper’s most severe
flaw. The paper is motivated by reducing the high dynamic memory of FO methods (e.g., storing
activations for backpropagation). The inner-loop update (Algorithm 1, Step 7) explicitly requires
a mini-batch FO gradient, ∇fIk (xsk). Computing this FO gradient requires backpropagation
and storing intermediate activations, leading to a dynamic memory cost of O(b ·P|al|). This is
the exact same dynamic memory cost as FO-SGD. The paper’s text (e.g., Section 4.3, Lines 325-
328: "thus do not need to store intermediate results", "dynamic memory only reaches maxl |xl|")
is factually incorrect and fundamentally misunderstands the cost of its own algorithm. The
data in the paper confirm this. Table 3 correctly lists VAMO’s dynamic memory as
Pl max{b ·|al|, |xl|}, identical to FO-SGD. Table 2 empirically shows VAMO’s memory (e.g., 21.46 GB) is
almost identical to FO-SGD’s (20.33 GB). The small increase is expected, as VAMO = FO-SGD
(dynamic) + |x| (snapshot state). The central claim that VAMO is a low-memory algorithm
(relative to FO-SGD) is false.

3. No Asymptotic Computational Advantage for O(1/T ) Convergence: To achieve the true
O(1/T ) convergence (i.e., remove the O(1/b) error), one must use the multi-point variant and set
q = d (Theorem 2).The computational complexity of VAMO (q = d) is O(qnS + (bd + bq)T) =
O(dnS + (bd + bd)T) = O(dnS + bdT ). Therefore, to achieve the same convergence rate as FOSVRG,
VAMO requires the identical asymptotic computational complexity. The paper’s claim
of computational efficiency is only valid when comparing the non-converging q = 1 version to the
converging FO-SVRG, which is an apples-to-oranges comparison.

4. Incorrect Computational Complexity in Table 1: The complexity for VAMO (q = 1) is
listed as O(nS +bdT ). This is incorrect. The inner loop (Step 7) computes two gradients: ∇fIk
(cost O(bd)) and ˆ∇fIk (cost O(bq), or O(b) for q = 1). The correct complexity is O(qnS +(bd+
bq)T). For q = 1, this is O(nS + (bd + b)T), which is O(nS + bdT ) only if d ≫ 1. This O(bqT)
term is missing from the table.

5. Empirical Evidence of Non-Convergence: In Figure 1b, VAMO(q = 1) clearly converges to
a final training loss that is visibly *higher* than that of FO-SGD. This plot empirically confirms
the O(1/b) noise ball limitation. This is a poor result, as it fails to match the solution quality of
the baseline FO-SGD, yet this is not discussed.

**Questions:**

1. The text in Section 4.3 (Lines 325-328) claims that VAMO does not need to store intermediate
activations. However, Step 7 of Algorithm 1 computes an FO gradient ∇fIk (xsk), which necessitates storing activations. Your own Table 3 and Table 2 confirm that VAMO has the same dynamic memory as FO-SGD. Can you please clarify this fundamental contradiction? Is the premise of the paper’s memory-saving benefits not incorrect?

2. The O(1/T + 1/b) rate converges to a noise ball, not a stationary point. Why is this
misleadingly presented as "similar to FO-SVRG" (which converges to 0) and an improvement
over FO-SGD (which also converges to 0)? Figure 1b seems to confirm that VAMO(q = 1) converges
to a worse solution.

3. The inner loop (Step 7) computes both an FO-grad (cost O(bd)) and a ZO-grad (cost O(bq)).
Why is the O(bqT) term missing from the complexity analysis in Table 1?

4. Given that VAMO(q = d) has the same rate and computation as FO-SVRG, why was this
Comparison (which is the only fair comparison of two O(1/T ) methods) is not included in the
experiments?

5. How sensitive are convergence and stability to the choice of α, μ, and q? Could you provide
empirical ablations or adaptive scheduling strategies, validating these settings across different
model dimensions?

**Details Of Ethics Concerns:**

This paper introduces a technically interesting and novel hybrid gradient estimator. The theoretical
analysis (deriving a dimension-independent O(1/T + ...) rate) is a non-trivial contribution.
However, the paper is built on a foundation of fundamentally incorrect and contradictory claims regarding
its practical advantages. The central premise that VAMO offers dynamic memory savings over
FO-SGD is demonstrably false, as refuted by the paper’s own algorithm, theoretical memory table,
and empirical memory measurements. The claim of computational superiority over FO-SVRG is also
illusory, as VAMO must match FO-SVRG’s asymptotic complexity (q = d) to match its O(1/T ) convergence
rate. In addition, the q = 1 variant is simply a method that converges faster than FO-SGD
to a worse solution (a noise ball of size O(1/b)), all while having the same dynamic memory footprint.
This is not the breakthrough claimed.

Because the paper’s central claims of computational and memory efficiency are not supported by (and
are, in fact, refuted by a correct analysis of the proposed algorithm, the work in its current form does
not meet the standards for publication at ICLR. The paper would need to be fundamentally reframed
to be honest about its actual (and much more modest) contributions: namely, a trade-off between
convergence speed and a final non-vanishing error.

---

> ### Author Response · Authors · 2025-11-19
>
> We thank the reviewer for their comments. However, the review contains several **factually incorrect statements regarding our memory analysis and experimental results**, which we clarify below.
>
> ---
>
> ### **Clarification on Memory Analysis**
>
> The reviewer states that our "central claim is that VAMO is a low-memory algorithm relative to FO-SGD." **This is factually incorrect.** We never claim that point in the paper. We explicitly state in Table 2 of our manuscript that VAMO’s dynamic memory is higher than FO-SGD, not lower. The reviewer appears to misunderstand the core motivation of VAMO:
>
> - **Memory Comparison**: VAMO is designed to be memory-efficient compared to **FO-SVRG** (which requires full FO gradients) and **Adaptive Methods like FO-Adam/FO-Adagrad** (which store momentum states). As shown in Table 1, VAMO's peak memory cost is 32% (RoBERTa-Large) and 26% (GPT-2 Medium) lower than FO-Adam on average. Compared to FO-Adagrad, VAMO's cost is 23% (RoBERTa-Large) and 20% (GPT-2 Medium) lower on average.
>
>
> - **“Intermediate Results” in Memory Analysis:** The text claiming VAMO "does not need to store intermediate results" (Lines 325-328) refers specifically to the **snapshot phase** (computing the full ZO gradient), *not* the inner-loop backpropagation for first-order gradients computation. Standard FO-SVRG requires expensive storage or re-computation at the snapshot; VAMO bypasses this via ZO estimation.
> We wish to point out the difference between the memory costs of the *snapshot* (where VAMO excels) and the *inner loop* (where VAMO matches FO-SGD). This is very important for understanding the memory analysis in the paper.
>
> **Table 1: Peak GPU Memory Consumption (in GB)**
>
> | Method | RoBERTa-Large (bs = 16) | RoBERTa-Large (bs = 32) | RoBERTa-Large (bs = 64) | GPT-2-Medium (bs = 16) | GPT-2-Medium (bs = 32) | GPT-2-Medium (bs = 64) |
> | --- | --- | --- | --- | --- | --- | --- |
> | FO-SGD | 4.63 | 6.72 | 10.83 | 7.08 | 11.50 | 20.33 |
> | FO-Adagrad | 7.21 | 10.46 | 16.54 | 10.25 | 15.74 | 27.57 |
> | FO-Adam | 8.62 | 11.59 | 17.66 | 11.59 | 17.05 | 29.07 |
> | **VAMO($q=1$)** | **5.95** | **7.83** | **11.96** | **8.40** | **12.62** | **21.46** |
>
> ### **Factual Error Regarding Convergence (Figure 1b)**
>
> We respectfully point out a **critical factual error** in the reviewer’s reading of our results. The reviewer claims: *"In Figure 1b, VAMO(q = 1) clearly converges to a final training loss that is visibly higher than that of FO-SGD."*
>
> - However, this is the opposite of what the figure shows. The blue line (higher loss) represents the convergence of FO-SGD, not VAMO.  The real conclusion is that VAMO converges to a **lower** loss and faster than FO-SGD in all experimental cases. The reviewer’s concern about "non-convergence" appears to stem from reading the plot backwards.
> - In addition, we must clarify that characterizing VAMO as "non-convergence" based on the $O(1/b)$ error term is a misunderstanding. The existence of a non-vanishing error term does not imply the algorithm fails to converge. Instead, this is a standard result indicating that VAMO converges to a neighborhood of a stationary point.
> - In practical LLM training with limited compute budgets, we cannot wait for FO-SGD to asymptotically converge to $\epsilon=0$. As demonstrated empirically, VAMO’s $O(1/T)$ rate dominates the early-to-mid training phases, providing a better solution than FO-SGD’s $O(1/\sqrt{T})$ within realistic time constraints.
>
> ### **Clarification on Computational Complexity & "Apples-to-Oranges" Comparison**
>
> Thank you for pointing that out. Regarding the table, we acknowledge the omission of the minor $O(bqT)$ inner-loop cost and have updated it. our formulation of $O(qnS + (bd+bq)T)$ is correct. For a fair comparison, FO-SVRG's inner loop is $O(2bdT)$ (total: $O(dnS + 2bdT)$) and ZO-SVRG's is $O(2bqT)$ (total: $O(qnS + 2bqT)$).
>
> However, we respectfully disagree with the reviewer’s critique that we must compare VAMO ($q=d$) to FO-SVRG, otherwise it is a "Apples-to-Oranges" Comparison. Requiring $q=d$ forces VAMO into a worst-case scenario that defeats the purpose of the algorithm. The core motivation of VAMO is precisely that we *can* achieve the flexible balance between computational efficiency and convergence rate, which FO-SVRG fails to achieve. In low dimensional problems (non-convex least square experiment in Section 6), we can use larger $q$ value for better convergence rate, as shown in Figure 1a. But in large model fine-tuning, running FO-SVRG (or VAMO with $q=d$) is computationally impractical. That is why we do not use the VAMO variant (q=d) in large models fine-tuning experiments (Section 6). We compare VAMO ($q=1$) against practical baselines (SGD, FO-Adam) to demonstrate that we can achieve SVRG-like acceleration without SVRG-like costs. This is not an "unfair" comparison; it is the **intended practical application** of the method. The choice represents the flexibility advantage of VAMO.

---

> > ### Author Response · Authors · 2025-11-19
> >
> > ### **Ablation Experiments**
> >
> > Regarding ablation experiments, we refer the reviewer to Appendix K, which contains detailed ablation studies on the mixing coefficient $\alpha$ and inner-loop steps $m$. Acknowledging the importance of such studies, we have further added ablations for $\mu$ in the revised manuscript to ensure completeness. We put the table of final training loss with varying smoothing parameter $\mu$, in fine-tuning GPT-2 on the MNLI dataset, as follows. We find that for a sufficiently small values of $\mu$ we see no noticeable difference in convergence rate. As we said in the paper, compared with ZO methods, the smoothing parameter in VAMO is less restrictive. And we chose $\mu=1e-3$ in the main experiment (Section 6).
> >
> > **Table 2: Final Training loss with Varying $\mu$**
> > | Algorithm (μ) | Loss at Step 5000 |
> > | :--- | :--- |
> > | $μ= 1e-05$ | 1.08 |
> > | $μ= 0.0001$ | 1.09|
> > | $μ= 0.001$ | 1.03 |
> > | $μ= 0.01$ | 1.04 |

---

### Official Review · Reviewer_P2tL · 2025-10-31

**Soundness:** 2
**Presentation:** 3
**Contribution:** 2
**Rating:** 6
**Confidence:** 3

**Summary:**

The research presents VAMO as a hybrid optimizer that works with big models under memory restrictions. The optimizer uses First-Order (FO) speed together with Zeroth-Order (ZO) efficiency by substituting the expensive SVRG variance-reduction algorithm step with a low-cost ZO gradient estimate. The algorithm provides fast dimension-independent convergence speed that outperforms SGD yet requires memory levels similar to SGD and less than Adam. The empirical results show VAMO runs at a slower pace than Adam but provides an attractive trade-off between performance and memory usage for restricted resource scenarios.

**Strengths:**

1. VAMO achieves a fast, linear convergence rate of $\mathcal{O}(1/T)$, which is an improvement over the $\mathcal{O}(1/\sqrt{T})$ rate of standard SGD.
2. A key advantage is that its convergence rate is independent of the model's parameter dimension $d$. This allows it to overcome the "curse of dimensionality" that makes purely Zeroth-Order (ZO) methods impractical for large models.
3. This paper provides a strong theoretical guarantee. VAMO's gradient estimator is designed to be unbiased.

**Weaknesses:**

1. The experimental results demonstrate that VAMO achieves better performance than SGD, but it fails to match the convergence speed and training efficiency of the Adam optimizer during large-scale fine-tuning.
2. The theoretical analysis shows that VAMO's convergence rate, while faster than SGD's, includes an additional error term of $\mathcal{O}(1/b)$ that is not present in the purely First-Order FO-SVRG algorithm.
3. The paper introduces a multi-point variant to minimize the additional error term. The improvement requires additional computational resources, which increase with the number of query points $q$, thus users need to decide between faster convergence and higher processing expenses.

**Questions:**

See weakness.

---

> ### Author Response · Authors · 2025-11-19
>
> We thank the reviewer for the constructive comments. We provide our feedback as follows.
>
> ---
>
> ### Test Accuracy and Memory Efficiency Compared with FO-Adam
>
> We acknowledge the reviewer's observation that FO-Adam generally achieves faster wall-clock convergence in terms of training loss. However, **VAMO is designed to solve the memory bottleneck, not to beat FO-Adam's speed.** Both our theoretical analysis and empirical experiments (see Table 1 below) show VAMO maintains a significantly lower peak memory footprint than FO-Adam. As shown below, VAMO's peak memory cost is 32% (RoBERTa-Large) and 26% (GPT-2 Medium) lower than FO-Adam on average. Compared to FO-Adagrad, VAMO's cost is 23% (RoBERTa-Large) and 20% (GPT-2 Medium) lower on average.
>
> More importantly, while FO-Adam may converge faster on the *loss*, this does not always translate to superior *generalization*. To explain it more clearly, we now explicitly present the final test accuracies and test losses (in parentheses) for the "Fine-Tuning Experiments" (from Section 6) in the table below (Table2). These results correspond to the fine-tuning of RoBERTa-Large, GPT-2 Medium, and GPT-2 Small on the MNLI and SST-2 datasets. For better comparison, we also present the peak memory usage for fine-tuning the RoBERTa-Large and GPT-2 Medium models (Table 2, Section 6) from our third experiment.
>
> As shown in Table 2, although VAMO converges slower than FO-Adam, it demonstrates better generalization ability in some tasks. For instance, VAMO's test accuracy is 8% higher than that of FO-Adam, in the RoBERTa-Large and 26% higher in the GPT-2 Medium (MNLI) task, while also achieving lower test losses in both. Besides, VAMO has 6% and 13% higher accuracy than that of FO-SGD and FO-Adagrad on average across all tasks, respectively.
> Compared with ZO methods, VAMO's test accuracy is 11% higher than that of ZO-SGD and ZO-SVRG on average.
>
> We believe the improved generalization ability might come from the use of full-batch gradients, since all other algorithms only use mini batch gradients, which may result in insufficient training. We plan to further investigate this phenonmenon in future works.
>
>
> **Table 1: peak GPU memory consumption (in GB) when fine-tuning RoBERTa-Large and GPT-2 Medium models.**
>
> | Method | RoBERTa-Large (bs = 16) | RoBERTa-Large (bs = 32) | RoBERTa-Large (bs = 64) | GPT-2-Medium (bs = 16) | GPT-2-Medium (bs = 32) | GPT-2-Medium (bs = 64) |
> | --- | --- | --- | --- | --- | --- | --- |
> | FO-SGD | 4.63 | 6.72 | 10.83 | 7.08 | 11.50 | 20.33 |
> | FO-Adagrad | 7.21 | 10.46 | 16.54 | 10.25 | 15.74 | 27.57 |
> | FO-Adam | 8.62 | 11.59 | 17.66 | 11.59 | 17.05 | 29.07 |
> | **VAMO($q=1$)** | **5.95** | **7.83** | **11.96** | **8.40** | **12.62** | **21.46** |
>
> **Table 2: Test Accuracy and Loss**
>
> | Model (Dataset) | FO-SGD | FO-Adagrad | FO-Adam | MeZO | ZO-SVRG | **VAMO (Ours)** |
> | --- | --- | --- | --- | --- | --- | --- |
> | **RoBERTa-Large** (MNLI) | 75 (0.67) | 72 (0.74) | 61 (0.98) | 73 (0.99) | 70 (1.01) | 78 (0.56) |
> | **GPT-2 Medium** (MNLI) | 60 (0.90) | 53 (1.33) | 48 (1.03) | 66 (0.89) | 69 (0.88) | 67 (0.85) |
> | **GPT-2 Small** (SST-2) | 90 (0.42) | 88 (0.37) | 98 (0.03) | 84 (0.59) | 72 (0.64) | 94 (0.16) |
>
>
>
> ---
>
> ### $O(1/b)$ Error Term in Convergence Analysis
>
> We acknowledge that VAMO ($q=1$) includes an additional $\mathcal{O}(1/b)$ error term compared to FO-SVRG. However, the core motivation of VAMO is to enable efficient variance reduction for large-scale optimization in resource-limited settings, a regime where standard FO-SVRG is often impractical due to prohibitive memory or computational cost. Since modern training of large machine learning models such as LLM require a staggering number of iterations (T), we believe the improvement of $\mathcal{O}(1/\sqrt{T})$ to $\mathcal{O}(1/T)$ compared to plain SGD represents a speedup that easily offsets the effects of $\mathcal{O}(1/b)$. This is especially true when the lower memory cost of VAMO allows for the choice of a larger $b$ in the FO component. Moreover, even with the additional $\mathcal{O}(1/b)$ factor, our experiments show that VAMO still achieves lower overall losses than most other optimizers (see Table 2), as explained before.

---

### Official Review · Reviewer_mfXT · 2025-11-01

**Soundness:** 3
**Presentation:** 3
**Contribution:** 2
**Rating:** 4
**Confidence:** 4

**Summary:**

The paper proposes VAMO, a hybrid variance-reduced optimizer that blends a mini-batch first-order (FO) gradient with a zeroth-order (ZO) SVRG-style correction computed at snapshot points. The key idea is to replace the expensive full-batch FO snapshot gradient with a full-batch ZO estimate and to weight the correction with a mixing coefficient \alpha. The authors prove a  convergence rate of O(1/T+1/b) independent of dimension d. Experiments span a synthetic task, MNIST MLP, and fine-tuning GPT-2 / GPT-2-Medium / RoBERTa-Large on SST-2 and MNLI, with GPU memory comparisons versus FO-SGD/Adagrad/Adam.

**Strengths:**

1. I like the discussion of memory: the paper argues VAMO’s snapshot uses forward-only ZO passes, so peak dynamic memory resembles ZO-SGD rather than FO-SVRG, and optimizer state is lighter than Adam/Adagrad. Included tables and a clear decomposition (weights / states / dynamics) are helpful.
2. The main bound removes the typical d dependence of ZO methods and improves over FO-SGD’s O(1/\sqrt{T}) in theory.

**Weaknesses:**

1.ZO methods inherently trade off performance, memory, and wall-clock. The paper treats ZO snapshots as “cheap,” but a full-batch snapshot still requires multiple forward passes per direction over the entire dataset (or many mini-batches). In practice, if memory is the bottleneck then ZO can help. However, runtime can balloon unless the number of directions q is tiny—yet shrinking q raises estimator variance and hurts convergence. Therefore, without an accounting of function evaluations (FEs) and throughput (e.g., tokens/sec), it’s unclear when VAMO is actually preferable to tuned FO baselines (Adam/Lion/Adafactor) or to ZO-SVRG variants with different q.

2. ZO can also be performed when the random direction is Gaussian, i.e., let u(x;\theta) = E_{\delta\sim N(0,\sigma^2 I) f(x+\delta;\theta), then \nabla_x u(x;\theta) = E_{\delta\sim N(0,\sigma^2 I) [\delta/\sigma^2  f(x+\delta;\theta). In high dimension, Gaussian vs. coordinate-wise directions can yield different estimator variance and smoothing bias. Yet this is not compared in the paper.

3. Fine-tuning results present training loss (showing convergence) and memory costs. However, I would expect also the task metrics (accuracy/F1 for SST-2/MNLI) and stability stats (divergence/NaNs) to show the effectiveness of VAMO.

4. SVRG-style methods hinge on the outer-loop frequency S and inner length m. The paper would benefit from an empirical study of how often to recompute ZO snapshots (and with what q) under a fixed compute budget; otherwise, it’s hard to see when VAMO is preferable to carefully tuned FO-SGD/Adam or existing ZO-SVRG variants.

**Questions:**

see weakenesses.

---

> ### Author Response · Authors · 2025-11-19
>
> We thank the reviewer for the insightful comments. We provide our feedback as follows.
>
> ---
>
> ### Clarification on Computation Efficiency of VAMO
> We acknowledge your concern, but we believe wall-clock time is a more common and practical choice compared with Function Evaluations (FEs). There is a distinction between the computational profiles of FO methods and ZO methods. Though plotting training loss versus function queries/evaluations is standard in pure Zeroth-Order baselines, it does not extend naturally to the hybrid setting we analyze in VAMO. Unlike pure ZO methods, VAMO requires both function evaluations and backpropagation. It is difficult to quantify the computational costs of FEs and back-propagation uniformly. Therefore, simply using FEs is an **unfair** comparison, as it fails to capture the dominant computational cost of the backpropagation process. This leads to the use of wall-clock time comparison, as it provides a more universal and uniform baseline for comparing algorithms of varying designs.
>
> Regrading your concerns about  the outer-loop frequency $S$ and inner length $m$, we hope to point out that ablation experiments on the parameter $m$ are already presented in **Appendix K (Figure 3a)**. We fine-tune GPT-2 on the MNLI dataset with varying $m$. The results show that a value of $m$ that is too large (e.g., $m>20$) leads to non-convergence in VAMO, while a value that is too small results in a large computational costs. Therefore, we choose $m=10$ in our main experiment (Section 6). We believe our wall-clock time plots in **Figure 2 (Section 6)** provide exactly the requested "fixed budget" analysis, showing VAMO is preferable to FO-SGD within a given time.
>
> ---
>
> ### Task Metrics and Stability Stats
>
> We agree with the reviewer that additional metrics such as test accuracy could further showcase the strength of VAMO. Therefore we now explicitly present the final test accuracies and test losses (in parentheses) for the "Fine-Tuning Experiments" (from Section 6) in the table below. These results correspond to the fine-tuning of RoBERTa-Large, GPT-2 Medium, and GPT-2 Small on the MNLI and SST-2 datasets. For better comparison, we also present the peak memory usage for fine-tuning the RoBERTa-Large and GPT-2 Medium models (Table 2, Section 6) from our third experiment.
>
> - **Test Accuracy Analysis:** As shown in the table, although VAMO converges slower than FO-Adam, it demonstrates better generalization ability in most tasks. For instance, VAMO's test accuracy is 8% higher than that of FO-Adam in RoBERTa-Large experiments and 26% higher in the GPT-2 Medium (MNLI) experiments, while also achieving lower test losses in both. Meanwhile, the peak memory cost of VAMO is 32% and 26% lower than that of FO-Adam in the RoBERTa-Large and GPT-2 Medium tasks, respectively. Compared with other algorithms, VAMO not only outperforms them in wall-clock convergence (see Figure 2), it also achieves a ~10% higher accuracy score on average, demonstrating both superior convergence speed and better generalization with a light memory footprint. We believe the improved generalization ability might come from the use of full-batch gradients, since all other algorithms only use mini batch gradients, which may result in insufficient training. We plan to further investigate this phenonmenon in future works.
>
> - **Stability (Divergence/NaNs):** We agree that the ZO estimator introduces variance, which can cause instability if not properly managed. This is precisely why the mixing coefficient $\alpha$ is a critical component. Our ablation in Appendix K.2 (Fig 3b) shows that instability only occurs when $\alpha$ is set inappropriately high (e.g., $\alpha > 0.2$) and by selecting a suitable $\alpha$ (e.g. $\alpha < 0.1$), this variance is effectively controlled. We choose $\alpha < 0.05$ in our main experiment (Section 6) and we observed **no instances of divergence or NaNs** in any of the experiments reported in our main tables.
>
> **Table 1: Test Accuracy and Loss**
> | Model (Dataset) | FO-SGD | FO-Adagrad | FO-Adam | MeZO | ZO-SVRG | **VAMO (Ours)** |
> | --- | --- | --- | --- | --- | --- | --- |
> | **RoBERTa-Large** (MNLI) | 75 (0.67) | 72 (0.74) | 61 (0.98) | 73 (0.99) | 70 (1.01) | 78 (0.56) |
> | **GPT-2 Medium** (MNLI) | 60 (0.90) | 53 (1.33) | 48 (1.03) | 66 (0.89) | 69 (0.88) | 67 (0.85) |
> | **GPT-2 Small** (SST-2) | 90 (0.42) | 88 (0.37) | 98 (0.03) | 84 (0.59) | 72 (0.64) | 94 (0.16) |
>
> **Table 2: Peak GPU memory Consumption (in GB)**
>
> | Method | RoBERTa-Large (bs = 16) | RoBERTa-Large (bs = 32) | RoBERTa-Large (bs = 64) | GPT-2-Medium (bs = 16) | GPT-2-Medium (bs = 32) | GPT-2-Medium (bs = 64) |
> | --- | --- | --- | --- | --- | --- | --- |
> | FO-SGD | 4.63 | 6.72 | 10.83 | 7.08 | 11.50 | 20.33 |
> | FO-Adagrad | 7.21 | 10.46 | 16.54 | 10.25 | 15.74 | 27.57 |
> | FO-Adam | 8.62 | 11.59 | 17.66 | 11.59 | 17.05 | 29.07 |
> | **VAMO** | **5.95** | **7.83** | **11.96** | **8.40** | **12.62** | **21.46** |

---

> > ### Author Response · Authors · 2025-11-19
> >
> > ### Choice of Random Directions
> >
> > Thank you for your suggestion. We utilized uniform sampling on the unit sphere to maintain consistency with established ZO-SVRG baselines [1] and ensure a fair, direct comparison. And in standard ZO optimization literature, comparative evaluations of different random direction types are generally not the primary focus of experiments. Theoretical literature suggests that in high-dimensional settings, both uniform, Gaussian sampling directions and coordinate-wise directions  can achieve similar convergence rates ($O(d)$ dependence) [2]. Besides, the choice of estimators is not the point in this paper. Therefore we maintain the standard estimator to focus our contribution on the novel *hybrid optimization framework* of VAMO.
> >
> > ---
> >
> > [1] Gautam, Tanmay, et al. "Variance-reduced zeroth-order methods for fine-tuning language models." *arXiv preprint arXiv:2404.08080* (2024).
> >
> > [2] Liu, Sijia, et al. "A primer on zeroth-order optimization in signal processing and machine learning: Principals, recent advances, and applications." *IEEE Signal Processing Magazine* 37.5 (2020): 43-54.

---

### Note · Authors · 2025-12-30

I have read and agree with the venue's withdrawal policy on behalf of myself and my co-authors.